# Entanglement and quench dynamics in the thermally perturbed tricritical fixed point

**Csilla Király**[1,2,3⋆] **and Máté Lencsés** [3†]

**1** Department of Theoretical Physics, Institute of Physics, Budapest University of Technology and Economics, Műegyetem rkp. 3., H-1111 Budapest, Hungary
**2** BME-MTA Statistical Field Theory 'Lendület' Research Group, Budapest University of Technology and Economics, Műegyetem rkp. 3., H-1111 Budapest, Hungary
**3** HUN-REN Wigner Research Centre for Physics , Konkoly-Thege Miklós u. 29-33, 1121 Budapest, Hungary

⋆ kiraly.csilla@wigner.hun-ren.hu , † lencses.mate@wigner.hun-ren.hu

## Abstract

We consider the Blume–Capel model in the scaling limit to realize the thermal perturbation of the tricritical Ising fixed point. We develop a numerical scaling limit extrapolation for one-point functions and Rényi entropies in the ground state. In a mass quench scenario, we found long-lived oscillations despite the absence of explicit spin-flip symmetry breaking or confining potential. We construct form factors of branch-point twist fields in the paramagnetic phase. In the scaling limit of small quenches, we verify form factor predictions for the energy density and leading magnetic field using the dynamics of one-point functions, and branch-point twist fields using the dynamics of Rényi entropies.

# 1  Introduction

Over the past few decades, the non-equilibrium dynamics of quantum systems has been at the forefront of fundamental physics research. Protocols in one-dimensional many-body systems realizing unitary quantum dynamics, such as the quantum quench scenario, have been utilized to study relaxation and thermalization hypotheses [1,2] both in experimental [3–6] and theoretical setups such as in [7,8], see also [9] for an extensive review.

Entanglement dynamics is a powerful probe of thermalization. In certain $1 + 1$ dimensional systems, linear-in-time growth of entanglement due to quasi-particle pairs [10–12] signifies exponential relaxation of observables and thermalization. Nevertheless, there exist one-dimensional quantum systems that fail to relax after a quench [13–15], a fact that was experimentally also observed [16,17]. Such behavior is usually attributed to integrability breaking and confinement. However, long-lived oscillations and the suppression of linear growth were also observed after a mass quench in the integrable $E_8$ scattering theory, realized by the scaling limit of the quantum critical transverse field Ising model in longitudinal field [18].

In this paper, we study an integrable quantum field theory, namely the $E_7$ model, the thermal perturbation of the tricritical Ising fixed point [19,20]. In contrast to $E_8$, this model possesses a $\mathbb{Z}_2$ spin-flip symmetry and two phases related to its spontaneous symmetry breaking. The paramagnetic or disordered phase hosts $\mathbb{Z}_2$ even/odd particles, while the ferromagnetic or ordered phase has two-fold degenerate vacua, odd kinks interpolating among them and even particles living above the vacua, which can be represented as different two-kink bound states. Similarly to the $E_8$ case, we realize quantum quenches in a spin chain realization of the model, namely the quantum Blume–Capel model [21–25] and found one-particle oscillations of one-point functions and Rényi entropies. Within the timescales accessible to our numerical simulations, the system exhibits persistent undamped oscillations, while the corresponding entropies do not show the linear growth characteristic of thermalization. This behavior is notable as it occurs in the absence of explicit spin-flip symmetry breaking or confining potentials.

In addition to the qualitative long-time behavior, we made a systematic scaling limit analysis of the spin chain realization of the $E_7$ model, which was put forward in the quantum Ising model and its scaling field theory in [18,26]. This process consists of several steps. Numerical results were obtained for the spin chain for various couplings using the infinite time evolving block decimation (iTEBD) algorithm [27,28]. In equilibrium, we verified the scal-

ing dimensions of local operators, while for the out-of-equilibrium dynamics, we extracted the mass-coupling relation and employed quench spectroscopy to obtain the theory's spectrum [15, 18, 29]. With these in hand, we were able to extrapolate the quench dynamics to the scaling field theory. This is a powerful method to study spatial entanglement properties in quantum field theories, where available numerical methods are rare and quite involved [30] or consider other partitions, such as chiral entanglement [31].

Such analysis shows quantitative agreement with quantum field theory results, most notably produces the expected mass ratios. Quench dynamics of one-point functions of the thermal and the order field agrees excellently with the perturbative results [32–34] given in terms of known form factors of local fields [35–37]. Moreover, we obtained results for von Neumann and Rényi entropies in terms of branch-point twist field form factors [38–41], that we computed in the paramagnetic phase.

In addition to the $\mathbb{Z}_2$ spin-flip symmetry, the trictitical Ising model has supersymmetry [42–44] and non-invertible symmetries [45–49]. The thermal perturbation breaks both, leaving only the spin-flip symmetry of the $E_7$ model. However, the high- and low-temperature phases are connected via the non-invertible Kramers–Wannier duality transformation. Indeed, our results also reflect the duality, as the order operator exhibits nontrivial behavior after quenches in the ferromagnetic phase. This is due to the fact that $\mathbb{Z}_2$ operators have non-vanishing form factors with even particle configurations only in the ferromagnetic phase [36]. Unfortunately, the Kramers–Wannier transformation and other non-invertible symmetries of the Blume–Capel model are not known explicitly. Such information would provide the exact tricritical point.

**Summary of our results** We verify that the scaling limit of the Blume–Capel spin chain with a certain set of couplings realizes the $E_7$ model. We carried out a detailed extrapolation of the iTEBD results for one-point functions of local operators and Rényi entropies. All the exponents are compatible with $E_7$ expectations. Moreover, in a dynamical setting we numerically obtained the mass-coupling relation on the Blume–Capel chain, and used mass quench spectroscopy to verify that the spectrum is consistent with the $\mathbb{Z}_2$ even part of $E_7$ mass spectrum. Using the extension of integrable bootstrap program, we obtained the branch-point twist field form factors up to two particles in the paramagnetic phase. We utilized a mass quench in the perturbative regime to further verify known form factor results for the thermal field and the magnetic field: scaling limit extrapolated iTEBD results match very well with the theoretical expectations for time evolution of one-point functions. We carried out the same analysis for the time evolution of Rényi entropies and found excellent agreement with perturbative results using the branch-point twist field form factors.

**Outline of the paper** In Section 2 we recall the necessary knowledge on the tricritical fixed point and its thermal deformation, and elaborate on the spin chain realization and its scaling limit. In Section 3 we summarize the form factor bootstrap and its extension to branch-point twist fields. We present our results on the even branch-point twist field form factors up to two particles in the paramagnetic phase. In Section 4 we recall the perturbative results for time evolution for small mass quenches in quantum field theories, then we present the agreement with matrix product state simulations in the scaling limit.

| Field | Name | wieghts | Kac-indices | $\eta$ | $N$ |
|-------|------|---------|-------------|--------|-----|
| $\mathbb{1}$ | identity | $(0,0)$ | $(1,1)$ or $(3,4)$ | $\mathbb{1}$ | $\mathbb{1}$ |
| $\sigma$ | leading magnetic field | $\left(\frac{3}{80},\frac{3}{80}\right)$ | $(2,2)$ or $(2,3)$ | $-\sigma$ | $\mu$ |
| $\epsilon$ | thermal | $\left(\frac{1}{10},\frac{1}{10}\right)$ | $(1,2)$ or $(3,3)$ | $\epsilon$ | $-\epsilon$ |
| $\sigma'$ | subleading magnetic | $\left(\frac{7}{16},\frac{7}{16}\right)$ | $(2,1)$ or $(2,4)$ | $-\sigma'$ | $\mu'$ |
| $t$ | vacancy/chemical potential | $\left(\frac{3}{5},\frac{3}{5}\right)$ | $(1,3)$ or $(3,2)$ | $t$ | $t$ |
| $\epsilon''$ | irrelevant | $\left(\frac{3}{2},\frac{3}{2}\right)$ | $(1,4)$ or $(3,1)$ | $\epsilon''$ | $-\epsilon''$ |

Table 1: Primary fields in the tricritical Ising model. In the last two columns we indicated the transformation properties under the $\mathbb{Z}_2$ spin-flip symmetry $\eta$ and the Kramers–Wannier duality $N$.

## 2 Tricritical Ising model and its thermal perturbation

### 2.1 Tricritical Ising CFT

The tricritical Ising model is described by the $\mathcal{M}_{4,5}$ minimal conformal field theory. This model has six primary fields, among which are the identity, four nontrivial relevant fields and an irrelevant one; we summarize the conformal dimensions and symmetry properties in Table 1.

The theory has a $\mathbb{Z}_2$ spin-flip symmetry, under which there are even and odd fields, as indicated in Table 1. As it turns out, the invertible $\mathbb{Z}_2$ symmetry is part of the total symmetry category, and it is accompanied with non-invertible symmetries. In minimal models the non-invertible symmetries are in one-to-one correspondence with topological lines, hence they can be characterized with the primary fields. The full symmetry is generated by $1, \eta, N, W, \eta W$ and $WN$ [49] lines.

The $\eta$ line corresponds to the $\mathbb{Z}_2$ spin-flip symmetry and $N$ is related to the Kramers–Wannier duality, which relates spin or order fields $\sigma/\sigma'$ to disorder fields $\mu/\mu'$. Transformation properties under these symmetries are also indicated in the last two columns of Table 1. It is easy to see that the $N$ transformation realizes the Kramers–Wannier duality between the low- and high-temperature phases of the thermal perturbation cf. Subsection 2.2.

One can then add the four relevant fields as perturbations to induce a renormalization group flow, defined by the action

$$S = S_{\text{TIM}} + \sum_i g_i \int d^2x \Phi_i(x), \tag{1}$$

where the sum runs over the relevant primary fields. Aspects of single perturbations were studied in [50]. Zamolodchikov studied the $t$ perturbations in [51,52]. Mulitple $\mathbb{Z}_2$ even perturbations were studied in [53]. Combinations of thermal and magnetic deformations were studied in the context of confinement and the decay of the false vacuum in [54,55] respectively. $\mathcal{PT}$ symmetric perturbations and the relation to multicritical Lee–Yang theories were studied in the recent series of works [56–58]. The role of non-invertible symmetries of the perturbed theories was considered recently in [59,60].

## 2.2   $E_7$ **model, spectrum and** $S$-**matrix**

In this paper, we consider the $E_7$ model, which is the thermal perturbation of the tricritical fixed point, given by the action:

$$\mathcal{S}_{E_7} = \mathcal{S}_{\text{TIM}} + \tilde{\lambda} \int d^2 x \, \epsilon(x) \tag{2}$$

with $\tilde{\lambda} > 0$ leading to the high-temperature phase and $\tilde{\lambda} < 0$ to the low-temperature phase. This perturbation only commutes with the invertible $\eta$ line [49]. The non-invertible $N$ line acts as a duality between the high- and low-temperature phases, known as Kramers–Wannier duality. The $\mathbb{Z}_2$ symmetry is preserved and remains unbroken in the high-temperature phase, while it is spontaneously broken in the low-temperature phase, leading to two degenerate ground states.

Moreover, this model is integrable [61], and the scattering is well-known [19, 20]. The theory has seven stable particles with masses given in Appendix A. Particle excitations are even or odd under the $\mathbb{Z}_2$ symmetry, the scattering is diagonal and the bound state structure reflects the $\mathbb{Z}_2$ symmetry. In the following, we briefly discuss some features in the two phases.

In the high-temperature phase $\tilde{\lambda} > 0$, the particles are excitations over the unique ground state and are even/odd under the $\mathbb{Z}_2$ symmetry. Even/odd fields have nonvanishing matrix elements between even/odd combinations of states, respectively. In particular, vacuum expectation values of odd fields (such as the magnetic and subleading magnetic fields) are zero [36, 62]. On the contrary, the disorder field (the Kramers–Wannier partner of the magnetization) has nonvanishing matrix elements in even configurations (see [36]). In the broken phase, i.e. $\tilde{\lambda} < 0$, there are two degenerate vacua, the excitations are either kinks interpolating between the vacua (odd particles), or bound states thereof (even particles). In this case the magnetization operator has nonvanishing matrix elements with even particles. This is the manifestation of the Kramers–Wannier duality, mapping between order and disorder fields.

Later we consider a mass quench scenario. We prepare the system to be in its ground state at some $\tilde{\lambda}$. Then at time $t = 0$ we suddenly change the coupling $\tilde{\lambda} \to \tilde{\lambda} + \delta_\lambda$ and let the system time evolve with the new Hamiltonian. The thermal field $\epsilon$ is an even field, therefore such a quench excites only even configurations, limiting the available spectrum to even particles, see Subsection 4 for a perturbative argument. This is reminiscent of the Ising model, where only particle pairs are created during the mass quench [10]. Similarly, in the ferromagnetic mass quench spectroscopy we do not expect to detect kink states, since the quench operator is even and thus unable to create kink excitations.

## 2.3   **The quantum Blume–Capel model**

The Blume–Capem model is a spin-1 quantum chain realizing the tricritical fixed point is given by the Hamiltonian [25]

$$H_{BC} = \xi \sum_i \left[ \alpha (S_i^x)^2 + \beta S_i^z + \gamma (S_i^z)^2 - S_i^x S_{i+1}^x \right] \tag{3}$$

where $S_i^{x,z}$ are three-by-three matrices acting on the spin at site $i$ given as

$$S_i^x = \frac{1}{\sqrt{2}} \begin{pmatrix} 0 & 1 & 0 \\ 1 & 0 & 1 \\ 0 & 1 & 0 \end{pmatrix}_i, \quad S_i^z = \begin{pmatrix} 1 & 0 & 0 \\ 0 & 0 & 0 \\ 0 & 0 & -1 \end{pmatrix}_i. \tag{4}$$

In [25] von Gehlen identified a line of tricritical points. In particular, for $\gamma = 0$, the other two parameters were estimated to be

$$\alpha_{\text{tc}} \approx 0.910207, \qquad \beta_{\text{tc}} \approx 0.415685 \tag{5}$$

and the thermal perturbation was identified as

$$H^\perp = \sum_{i=1}^{N} h_i^\perp = \sum_{i=1}^{N} \left( -\sin\theta (S_i^x)^2 + \cos\theta S_i^z \right),\tag{6}$$

with $\tan\theta \approx 2.224$, based on a finite-size scaling argument. This operator is manifestly $\mathbb{Z}_2$ symmetric, and it was found that depending on the sign of the coupling, this perturbation brings the model to the low- or high-temperature phase of the $E_7$ model, based on the spectrum of low-lying states.

In our numerical calculation we use this set of reference values to realize the $E_7$ model, i.e. we consider the following Hamiltonian:

$$H_{E7} = \xi \left\{ \sum_i \left[ \alpha_{\text{tc}}(S_i^x)^2 + \beta_{\text{tc}} S_i^z - S_i^x S_{i+1}^x \right] + \lambda H^\perp \right\},\tag{7}$$

where $\lambda < 0$ gives the ordered or ferromagnetic phase, and $\lambda > 0$ realizes the disordered or paramagnetic phase.

One might be tempted to identify the $\epsilon$ field of the CFT with the scaling limit of $h_i^\perp$. This claim eventually turns out to be false. In Section 2.4 we show that in the scaling limit this operator has a nonzero expectation value at the critical point, therefore it cannot be identified with a conformal primary field. Hereafter, we denote the "true" energy density operator by $\mathcal{E}_n$, which we suppose to have the general form:

$$\mathcal{E}_n = \varepsilon_x (S_n^x)^2 + \varepsilon_z S_n^z + \varepsilon_{x,x} S_n^x S_{n+1}^x,\tag{8}$$

which is manifestly $\mathbb{Z}_2$ even. Note that since in any case the coefficient of the nearest neighbor term can always be scaled out, the effect of $H^\perp$ as a perturbation will lead to the same off-critical theory.

A similar case arises in the transverse field Ising model. There, a natural first assumption is to identify the scaling limit of the transverse field with the energy density operator. On the other hand, deforming the nearest neighbor coupling would lead to the same effect, hence one has every right to call $\sigma_i^z \sigma_{i+1}^z$ the energy density operator. Moreover, both operators have nonzero expectation values in the quantum critical point of the chain [63], which prevents them from being primary fields. The correct choice is $\sigma_i^z \sigma_{i+1}^z - \sigma_i^x$, an operator even under spin flip and odd under the Kramers–Wannier duality transformation $\sigma_i^z \sigma_{i+1}^z \longleftrightarrow \sigma_i^x$ [64,65].

Note that to our knowledge the Kramers–Wannier duality transformation is not known for the Blume–Capel model, nevertheless, we keep with this analogy, and treat the results for the scaling limit of $h_i^\perp$ as an operator as it is, and allow for a constant expectation value in the tricritical point. We will see, that apart from the constant VEV, this operator has the right scaling dimension, i.e. form factor calculations are applicable.

There is no such trouble with the leading magnetization operator. It is identified with $S_i^x$. We found (see 2.4) that this operator indeed scales as expected and its expectation value tends to zero in the critical point.

## 2.4 Scaling limit

The Blume–Capel spin chain can be studied numerically using the iTEBD algorithm [27, 28]. The scaling limit can be achieved in the spirit of [18, 26], by tuning the thermal coupling $\lambda$ in (7) closer and closer to its critical value and appropriately rescaling the energy scale, staying on the line of constant physics. The critical point is at $\lambda = 0$, however, tuning $\lambda \to 0$ together with $\xi \to \infty$ is equivalent to decreasing the lattice spacing $a$. Finally, one can extrapolate to the scaling limit, which is supposed to be described by the $E_7$ scattering theory.

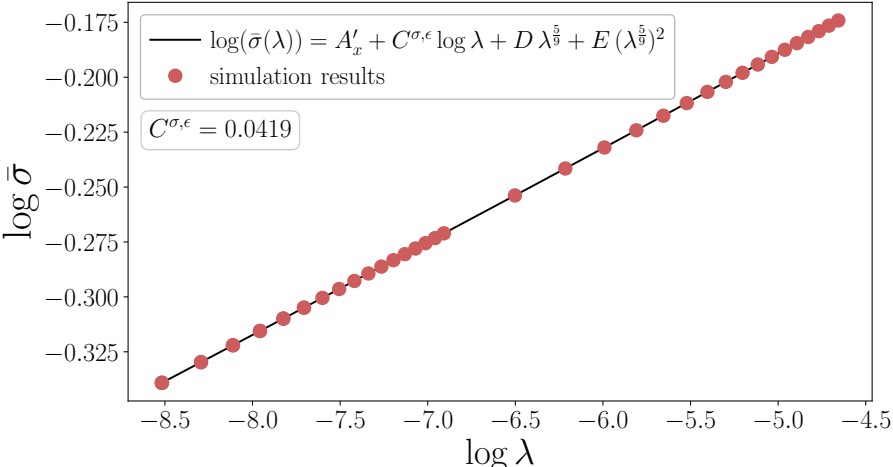

Figure 1: The scaling of the expectation value of the magnetic field. We found the expected scaling behavior, with the measured exponent 0.0419 in an excellent agreement to its theoretical value.

Based on the conformal dimensions in the CFT, we expect that scaling operators have the form:

$$\sigma(x = na) = a^{-\frac{3}{40}} \bar{s} S_n^x, \tag{9}$$

$$\epsilon(x = na) = a^{-\frac{1}{5}} \bar{e} \mathcal{E}_n, \tag{10}$$

where $\bar{s}$ and $\bar{e}$ are unknown constants relating the normalization of the scaling fields and spin chain opeartors[1]. In the scaling limit, the $E_7$ theory is given by the Hamiltonian:

$$H = H_{\text{CFT}} + \tilde{\lambda} \int dx \epsilon \tag{11}$$

where $\tilde{\lambda}$ has dimension $2 - 2h_\epsilon$ and is related to the chain coupling as

$$\tilde{\lambda} = \xi a^{-(1-2h_\epsilon)} \bar{e} \lambda. \tag{12}$$

The overall energy scale is set by $\xi$. On dimensional grounds, the mass scale in the off-critical theory can be written in terms of the couplings as:

$$m_\lambda = \kappa_l \xi \lambda^{\frac{1}{2-2h_\epsilon}}. \tag{13}$$

Expectation values of a local operators in scaling field theories i.e. perturbed conformal field theories can also be written in terms of the coupling. For the Hamiltonian (11) we have

$$\langle \mathcal{O} \rangle = \mathcal{A}_\mathcal{O} \tilde{\lambda}^{C^{\mathcal{O},\epsilon}}, \tag{14}$$

where $C^{\mathcal{O},\epsilon} = \frac{2h_\mathcal{O}}{2-2h_\epsilon}$. We expect the same scaling form to be true for the spin chain operators as well, so that

$$\langle S^x \rangle = A_x \lambda^{\frac{1}{24}} \tag{15}$$

$$\langle h^\perp \rangle = A_\perp \lambda^{\frac{1}{9}} + \text{const.}, \tag{16}$$

---

[1]In the transverse field Ising chain, the corresponding relation between the conformally normalized magnetization operator $\sigma$ and the and the Pauli $\sigma^z$ on the chain reads as $\sigma_i^z = a^{1/8} s \sigma(x = ia)$, where $s$ is known analytically, and is given in terms of Glaisher's constant [63, 66].

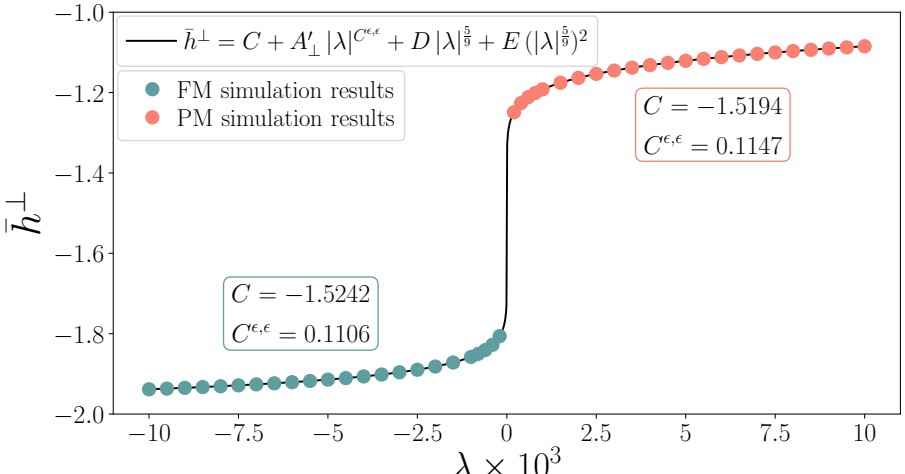

Figure 2: Expectation value of the perturbing operator in the low- and high-temperature phases, with $\lambda < 0$ and $\lambda > 0$ respectively, with scaling limit fits separately in both phases. One can see that it exponentially converges to approximately the same constant and with the same exponent in both phases.

where const. accounts for the finite expectation value in the critical point, discussed in the previous subsection. The $\mathcal{A}_{\mathcal{O}}$ coefficients are known in the $E_7$ field theory [62], but as it was mentioned earlier, their relation to the lattice values is unknown.

We made a systematic study using iTEBD to determine the scaling of these operators. We constructed the ground state for various couplings $|\lambda| \ll 1$ in (7). Then scaling limit extrapolations were done with the functional form:

$$A + C \log |\lambda| + D|\lambda|^{5/9x} + E(|\lambda|^{5/9x})^2, \tag{17}$$

where the first two terms account for the scaling field theory expectation of eq. (15), the last two incorporate the effect of finite lattice spacing cf. eq. (12). Here, $x = 1$ for one point functions of $\sigma$ and $\epsilon$, and $x = \frac{1}{n}$ for the $n$th Rényi entropy i.e. the logarithm of the branch-point twist field one-point function (cf. 3) to account for the unusual scaling in the lattice spacing [18, 26, 67], and we found that for the $n$th Rényi entropy $D = 0$. Note that this form assumes that expectation values in the critical point vanish, which is true for the magnetization and the branch-point twist fields. In the case of the $h^\perp$ operator, we fitted the exponential of the above function with the addition of a constant term.

Our results can be found in Figures 1 and 2. For the magnetic field the form given in (15) is verified, the expected power is fitted within 1% accuracy. However, as discussed in 2.3, the situation is slightly different for $h^\perp$. We found that the expectation value is nonzero in the critical point, however, the results exponentially converge to the finite expectation value at zero coupling, with an exponent within 4% error with the expected one. We collected the fit data in Table 2.

We carried out the same analysis for entropies i.e. the logarithm of branch-point twist field one-point functions (see Section 3). The results for the first four entropies match very well with the theoretical predictions, and are presented in Figure 3 and in Table 2.

Now let us comment on the energy scale $\xi$ in (7). In the above procedure it was always chosen to be 1, as it does not affect the value of one-point functions and the entanglement properties, since these depend only on the quantum state in hand. However, when dynamics of these quantities is considered, the value of $\xi$ sets the time scale. Nevertheless, in a time-dependent setting, the effect of $\xi$ can always be compensated by the rescaling of time. In our simulations this was done as follows.

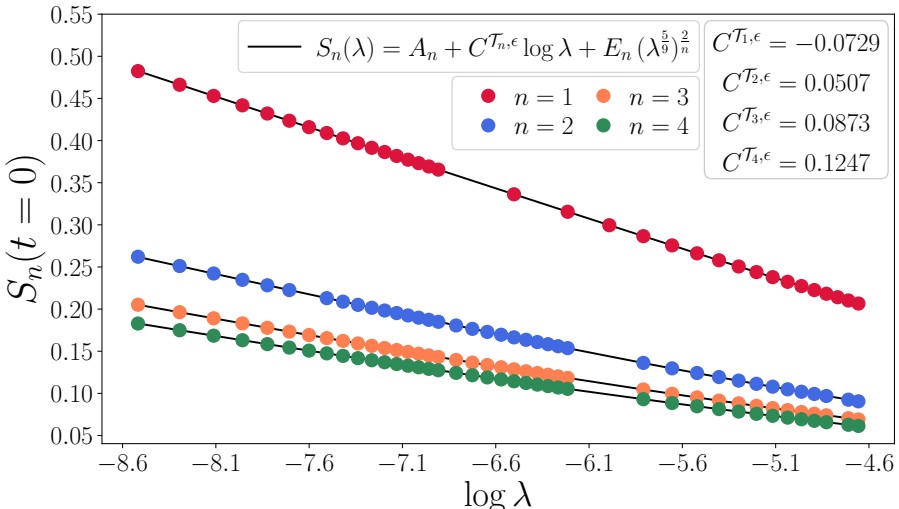

Figure 3: Scaling limit of the Rényi entropies. Unusual scaling ansatz is important to extract the conformal weights of the twist fields, which in turn are in excellent agreement with the theory.

| $\mathcal{O}$ | $C^{\mathcal{O},\mathcal{E}}$ theory | $C^{\mathcal{O},\mathcal{E}}$ fit |
|---|---|---|
| $\sigma$ FM | 0.0417 | 0.0419 |
| $\mathcal{E}$ FM | 0.1111 | 0.1106 |
| $\mathcal{E}$ PM | 0.1111 | 0.1147 |
| $\mathcal{T}_1$ | $-0.0648$ | $-0.0729$ |
| $\mathcal{T}_2$ | 0.0486 | 0.0507 |
| $\mathcal{T}_3$ | 0.0864 | 0.0873 |
| $\mathcal{T}_4$ | 0.1215 | 0.1247 |

Table 2: VEV exponents from field theory dimension counting vs. scaling limit fits from MPS computations.

Let us perform a quantum quench simulation from some $\lambda_0$ to $\lambda$, and measure the time dependence of certain one-point functions, e.g. of $S^x$. Such a time-dependent quantity has a dominant oscillatory behavior, controlled by the mass of the lightest particle which couples to the quench operator:

$$Q(t) \approx A + B\cos(m_\lambda t + \alpha),\tag{18}$$

where $m_\lambda$ can be fitted from simulation data for various quantities. This observation can be utilized in the following way. We perform a set of quenches with fixed $\xi$ and different $\lambda_0$, but with the same relative magnitude or quench parameter $q = \frac{\lambda - \lambda_0}{\lambda_0}$. Then we rescale the time, to reach the same period. This way we obtain $m_\lambda$ as a function of $\lambda$, which can then be fit with the expected mass-coupling. This is summarized in Figure 4 for the magnetization for $q = 0.05$, where we found that in the Blume–Capel scaling limit $m_1 \approx 14.207\,\lambda^{5/9}$. Note that in this case the lightest excited particle is $m_2$, and the rescaling in the middle and right panels of Figure 4 were done in a way that the leading oscillations have frequency $m_2/m_1 \approx 1.285$.

In addition, we performed another dynamical test of this claim by implementing a mass quench scenario, then studying the time evolution of certain one-point functions. The time-dependent signal can be used to extract spectral information, a method called quench spectroscopy [15, 18, 29]. In Figure 5 we show the Fourier transform of the one-point function of the magnetization operator in the ferromagnetic phase after a certain quench. It is easy to

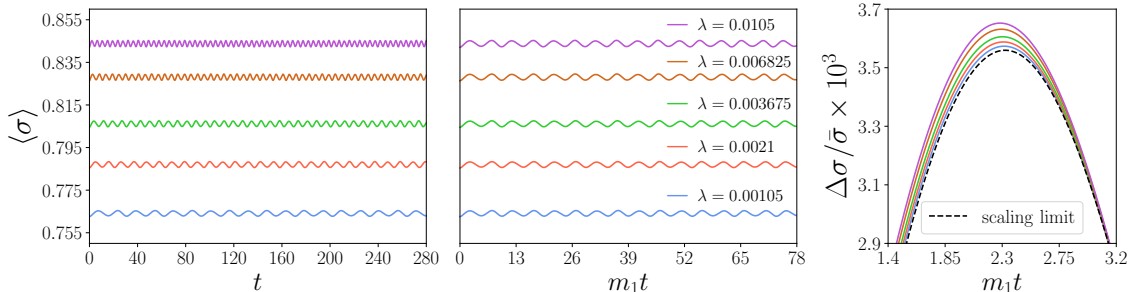

Figure 4: Rescaling of time/energy scale. In the left panel we show the data produced by iTEBD simulations for various quenches with 5% change in the coupling. The post-quench couplings are indicated in the middle panel, where we show the rescaled data. On the rescaled data, one can perform a time-dependent extrapolation to the scaling limit, as we show in the right panel.

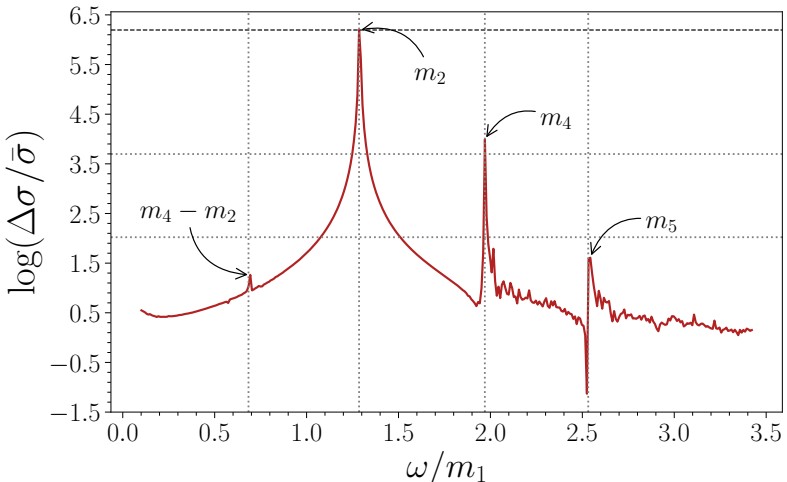

Figure 5: The power spectrum of the time evoultion of the magnetization after the mass quench with $\lambda_0 = 0.001 \rightarrow 0.00105$. The uppermost horizontal line is fitted by hand, the other two were computed using one-particle form factors from [35, 36]. Locations of peaks and the height of the particle peaks are compatible with the integrability results for even particles.

extract one-particle mass ratios, which are collected in Table 3. It is clear that these ratios are in excellent agreement with the expected *even* mass spectrum of the $E_7$ model. Therefore, the particle interpretation is also consistent with the $\mathbb{Z}_2$ parity of particles summarized in Table 9 given in Appendix A. Moreover, based on the perturbation theory presented in Subsection 4.1, the height of the peaks are related to the form factors of the quench operator and the operator in consideration, which is also consistent with our resluts.

The findings of this subsection provide further verification of von Gehlen's claim, that this perturbation approximates very well the $E_7$ model. However it would be extremely interesting to study the Kramers–Wannier duality in the context of this model. This would provide the exact location of the tricritical point, i.e. the fixed point of the duality transformation and the exact expression for the thermal field. This is out of the scope of the present paper.

| $r_i$ | theory | quench spectroscopy |
|---|---|---|
| $r_2$ | 1.286 | 1.286 |
| $r_4$ | 1.97 | 1.97 |
| $r_5$ | 2.532 | 2.543 |
| $r_4 - r_2$ | 0.684 | 0.694 |

Table 3: Mass ratios $r_i = \frac{m_i}{m_1}$ of one-particle states, extracted from quench spectroscopy of one-point functions. The results are compatible with the *even* particle series.

## 3 Branch-point twist field form factor bootstrap

### 3.1 Replica trick and branch-point twist fields

Entanglement properties in pure quantum states can be characterized by the von Neumann and Rényi entropies. The density matrix of a pure state $|\Psi\rangle$ is given as:

$$\rho = |\Psi\rangle \langle\Psi| \, . \tag{19}$$

Let us consider a spatial bipartition of the space to a finite subsystem $A$ and the rest of the world $B$. Then the reduced density matrix $\rho_A$ of the subsystem is given by the partial trace over $B$:

$$\rho_A = \text{Tr}_B \rho \, . \tag{20}$$

Then the $n$th Rényi entropy corresponding to this bipartition is

$$S_n^{AB} = \frac{1}{1-n} \log \text{Tr}_A \rho_A^n, \tag{21}$$

and the von Neumann entropy is

$$S^{AB} = -\text{Tr}_A \rho_A^n \log \rho_A^n = \lim_{n \to 1} S_n^{AB} \, . \tag{22}$$

These quantities are viable measures of quantum entanglement between the subsystems $A$ and $B$ in pure states.

In this paper, we are interested in the entanglement properties of 1+1 dimensional quantum field theories. A bipartition i given by endpoints of an interval, which can be chosen to $A = [0, x]$ without loss of generality. The trace of the $n$th power of the reduced density matrix in the ground state can be computed as a partition function over an $n$-sheeted Riemann surface [38], as illustrated in Figure 6 for $n = 3$.

Instead one can use the replica trick of [39]: trade the comlicated Riemann surface for $n$-copies of the original theory and introduce a $\mathbb{Z}_n$ symmetry and corresponding symmetry lines, representing the cuts of the Riemann surface. The symmetry lines terminate in the corresponding twist and anti-twist fields, which are known in general as branch-point twist fields. Then finally, the trace of powers of the reduced density matrix in the ground state can be calculated in terms of two-point functions of branch-point twist and anti-twist fields $\mathcal{T}_n$ and $\bar{\mathcal{T}}_n$,

$$\text{Tr}_A \rho_A^n \propto \langle \bar{\mathcal{T}}_n(x) \mathcal{T}_n(0) \rangle \, . \tag{23}$$

These objects can be computed in conformal field theories based on geometry and conformal symmetry arguments [39]. The computation results in the fact that the branch-point twist fields are primary fields of conformal dimension

$$\Delta^{\mathcal{T}_n} = \frac{c}{24}\left(n - \frac{1}{n}\right) \, . \tag{24}$$

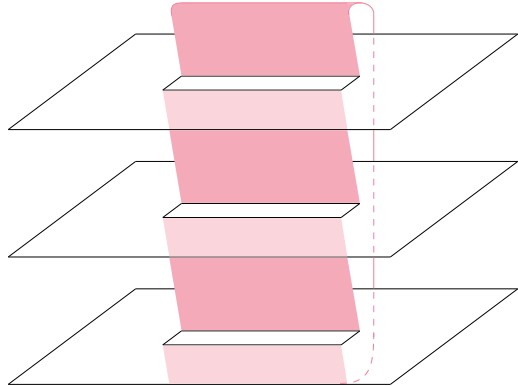

Figure 6: 3-sheeted Riemann surface

In a massive theory the conformal invariance is broken, and the geometric argument fails. However, the replica trick still proves to be useful [41]. A resolution of the identity can be inserted in the correlator (23) resulting in the usual form factor expansion.

From a clustering argument (see eg. [26]) it can be deduced, that in a massive theory for a large enough subsystem, the twist field two-point function is formally given as a square of the one-point function of the twist field. Taking the logarithm results in a factor of 2, reminiscent from the area law of entanglement [68]. Moreover, this argument was used in [18] to show that given a bipartition of an infinite system to two semi-infinite ones, the entanglement between these subsystems can be characterized by twist field one-point functions. We consider this situation in Section 4.

However, these quantities are not well-defined (they possess UV divergences), entanglement entropy differences lead to finite quantities. In particular, one can study a mass quench scenario, when oscillations in the Rényi entropies are related to one-particle states and the corresponding form factors. One of the goals of this paper is to determine these one-particle form factors in the $E_7$ scattering theory and verify them numerically using a mass quench in the quantum spin chain realization.

The symmetry of the replicated theory is $\mathbb{Z}_2^n \times \mathcal{S}_n$, since all copies have their own spin-flip ($\mathbb{Z}_2$ twist), and a priori there is no distinction between the $n$ copies; therefore, permuting them would not lead to any difference. However, considering twist and anti-twist field insertions, this symmetry reduces to $\mathbb{Z}_2 \times \mathbb{Z}_n$, since sheets are sewn together, therefore a spin-flip on one sheet would induce flipping in all sheets. Also, from the sewing, the permutation symmetry reduces to the cyclic subgroup.

At this point, we restrict ourselves to the paramagnetic phase, where there is only a single unique $\mathbb{Z}_2$ invariant vacuum. Since the branch-point twist fields are defined geometrically, depending only on the ground state configuration, we expect them to be even operators under $\mathbb{Z}_2$. This claim is further verified by our numerics: the time evolution of Rényi entropies after a mass quench is consistent with this picture, the oscillations that are present are compatible with the even sector of the $E_7$ spectrum.

The situation is different in the ferromagnetic phase, where there is a two-fold ground state degeneracy, and ground states transform nontrivially under the symmetry. However, different sectors are connected via a $\mathbb{Z}_2$ twist field (e.g. $\sigma$, the order parameter), and one has to consider composite twist fields. This was studied in the Ising model in [69]. In this paper we do not consider the ferromagnetic phase of the Blume–Capel model, however we are planning to revisit this issue.

## 3.2 Form factor bootstrap

In this section we recall the elements of the form factor bootstrap program [70–72] in integrable models.

Form factors are matrix elements of local operators. Integrability implies serious constraints on the dynamics of the theory, such as the purely elastic scattering of particles. This information is encoded in the so-called Feddeev–Zamolodchikov algebra:

$$A_a(\theta_1)A_b(\theta_2) = S_{ab}^{cd}(\theta_1 - \theta_2)A_d(\theta_2)A_c(\theta_1), \tag{25}$$

where the operators $A_a(\theta)$ are the Faddeev–Zamolodchikov operators, which create particles of type $a$ with rapidity $\theta$. From this point we consider only models with diagonal scattering i.e. there are no multiplets in the theory and during the scattering process, the particle types cannot change. For this reason, from now on we use the shortened notation $S_{ab} = S_{ab}^{ab}$ for the diagonal $S$-matrix element.

A multi-particle state is given as:

$$|\theta_1, \dots, \theta_n\rangle_{a_1, \dots, a_n} = A_{a_1}(\theta_1) \dots A_{a_n}(\theta_n)|0\rangle, \tag{26}$$

and a form factor of a local operator $\mathcal{O}$ is defined as the matrix element:

$$F_{a_1, a_2, \dots, a_n}^{\mathcal{O}}(\theta_1, \dots, \theta_n) = \langle 0| \mathcal{O}(0,0) |\theta_1, \dots, \theta_n\rangle_{a_1, \dots, a_n}. \tag{27}$$

Form factors satisfy the following properties:

**Monodromy**

$$F_{a_1, \dots, a_i, a_{i+1}, \dots a_n}^{\mathcal{O}}(\theta_1, \dots, \theta_i, \theta_{i+1}, \dots, \theta_n) = S_{a_i, a_{i+1}}(\theta_i - \theta_{i+1})$$
$$\times F_{a_1, \dots, a_{i+1}, a_i, \dots a_n}^{\mathcal{O}}(\theta_1, \dots, \theta_{i+1}, \theta_i, \dots, \theta_n) \tag{28}$$

$$F_{a_1, a_2, \dots, a_n}^{\mathcal{O}}(\theta_1 + 2\pi i, \theta_2, \dots, \theta_n) = F_{a_2, a_3, \dots, a_n, a_1}^{\mathcal{O}}(\theta_2, \theta_3, \dots, \theta_n, \theta_1) \tag{29}$$

**Kinematical pole**

$$-i \lim_{\tilde{\theta} \to \theta} (\tilde{\theta} - \theta) F_{\bar{a}, a, a_1, \dots, a_n}^{\mathcal{O}}(\tilde{\theta} + i\pi, \theta, \theta_1, \dots, \theta_n) = \left(1 - e^{2\pi i \omega_a} \prod_{j=1}^{n} S_{a, a_j}(\theta - \theta_j)\right) \tag{30}$$
$$\times F_{a_1, \dots, a_n}^{\mathcal{O}}(\theta_1, \dots, \theta_n), \tag{31}$$

where $\omega_a$ is the mutual semi-locality index of the operator $\mathcal{O}$ with respect to the particle of type $a$, and $\bar{a}$ denotes $a$'s charge conjugate.

**Bound state poles, higher bound state poles** Simple poles in the $S$-matrix at $\theta = iu_{ab}^c$ correspond to bound state poles associated with fusion processes $A_a \times A_b \to A_c$, where the $S$-matrix residue is $i(\Gamma_{ab}^c)^2$, leading to recursive relations for form factors:

$$-i \lim_{\theta_{ab} \to iu_{ab}^c} (\theta_{ab} - iu_{ab}^c) F_{a, b, a_1, \dots, a_n}^{\mathcal{O}}(\theta_a, \theta_b, \theta_1, \dots, \theta_n) = \Gamma_{ab}^c F_{c, a_1, \dots, a_n}^{\mathcal{O}}(\theta_c, \theta_1, \dots, \theta_n), \tag{32}$$

where $\theta_c = (\theta_a \bar{u}_{bc}^a + \theta_b \bar{u}_{ac}^b)/u_{ab}^c$ with $\bar{u} = \pi - u$.

The $S$-matrices can have higher order poles that are related to Coleman–Thun diagrams. These poles represent themselves as first order poles in the form factors. For a double pole in the $S$-matrix at angle $\phi = u_{ad}^c + u_{bd}^e - \pi$ the form factor has a simple pole at $\theta = i\phi$:

$$-i \lim_{\theta_{ab} \to i\phi} (\theta_{ab} - i\phi) F_{a, b, a_1, \dots, a_n}^{\mathcal{O}}(\theta_a, \theta_b, \theta_1, \dots, \theta_n) = \Gamma_{ad}^c \Gamma_{bd}^e F_{c, e, a_1, \dots, a_n}^{\mathcal{O}}(\theta + i\gamma, \theta, \theta_1, \dots, \theta_n), \tag{33}$$

where $\gamma = \pi - u_{cd}^a - u_{de}^b$.

**Clustering** The last property is the clustering or asymptotic factorization property [35, 73, 74],

$$\lim_{\alpha \to \infty} F_{r+l}^{\mathcal{O}_a}(\theta_1 + \alpha, \dots, \theta_r + \alpha, \theta_{r+1}, \dots, \theta_{r+l}) = \omega_a^{bc} F_r^{\mathcal{O}_b}(\theta_1, \dots, \theta_r) F_l^{\mathcal{O}_c}(\theta_{r+1}, \dots, \theta_{r+l}), \quad (34)$$

where $\mathcal{O}_a, \mathcal{O}_b, \mathcal{O}_c$ are fields with the same conformal dimension and $\omega_a^{bc}$ is a phase which depends on the phase conventions for the one-particle states.

### 3.3 Branch-point twist field form factor bootstrap

In this section, we summarize the extension of the form factor axioms to branch-point twist fields based on [41, 75].

The axioms above have to be modified for the case of the branch-point twist fields, $\mathcal{T}_n$ corresponding to $n$ copies. Now, a multiparticle state — and therefore also a form factor — carries copy or sheet indices: $\mu_i = (s_i, a_i)$ indicates an asymptotic particle of type $a_i$ on sheet $s_i$.

An important ingredient of the calculation is the vacuum expectation value of the brach-point twist field $\langle \mathcal{T}_n \rangle^{23}$. In general, there is no obvious way to extract twist field VEVs, or in any way relate it to the spin chain realization[4]. For this reason, we only consider quantities, which do not depend on VEVs , and in our form factor calculations we consider "normalized" form factors, meaning that

$$F_i^{\mathcal{T}_n} = \frac{\langle 0 | \mathcal{T}_n | i \rangle}{\langle \mathcal{T}_n \rangle}. \quad (35)$$

**Monodromy**

$$
\begin{aligned}
F_{\dots, \mu_i, \mu_{i+1}, \dots}^{\mathcal{T}_n}(\dots, \theta_i, \theta_{i+1}, \dots) &= \hat{S}_{\mu_i, \mu_{i+1}}(\theta_i - \theta_{i+1}) F_{\dots, \mu_{i+1}, \mu_i, \dots}^{\mathcal{T}_n}(\dots, \theta_{i+1}, \theta_i, \dots) \\
F_{\mu_1, \mu_2, \dots, \mu_n}^{\mathcal{T}_n}(\theta_1 + 2\pi i, \theta_2, \dots, \theta_n) &= F_{\mu_2, \mu_3, \dots, \mu_n, \hat{\mu}_1}^{\mathcal{T}_n}(\theta_2, \theta_3, \dots, \theta_n, \theta_1),
\end{aligned} \quad (36)
$$

where $\hat{\mu}_i = (s_i + 1, a_i)$ and we defined the multi-sheet $S$-matrix as:

$$\hat{S}_{\mu_i, \mu_{i+1}}(\theta_i - \theta_{i+1}) = (1 - \delta_{s_i, s_{i+1}}) + \delta_{s_i, s_{i+1}} S_{a_i, a_{i+1}}(\theta_i - \theta_{i+1}). \quad (37)$$

**Kinematical pole** The kinematical pole is also modified:

$$-i \lim_{\tilde{\theta} \to \theta} (\tilde{\theta} - \theta) F_{\bar{\mu}_a, \mu_b, \mu_1, \dots, \mu_n}^{\mathcal{T}_n}(\tilde{\theta} + i\pi, \theta, \theta_1, \dots, \theta_n) = \left[ \delta_{\mu_a, \mu_b} - (1 - \delta_{\mu_a, \mu_b}) \prod_{i=1}^{n} \hat{S}_{\mu_b \mu_i}(\theta - \theta_i) \right]$$
$$\times F_{\mu_1, \dots, \mu_n}^{\mathcal{T}_n}(\theta_1, \dots, \theta_n) \quad (38)$$

where $\bar{\mu}$ denotes the charge conjugation of the particle. In particular, the kinematical pole equation relates two-particle form factors to the twist field VEV, which in our calculation is set to be 1.

---

[2]Strictly speaking, VEV of the branch-point twist field is not a well-defined object, since a twist always comes in pair with an anti-twist. However, one can include twisted sectors, such as in the case of ordinary global discrete symmetries. This notion is also important from the point of view of the cluster equation (39).

[3]From now on, expectation values and matrix element are meant to be between states in the replicated theory.

[4]The only exception known to us are Yang–Baxter integrable models, where one can relate the reduced density matrix to the so-called corner transfer matrix [76].

**Bound state poles** Bound state poles only occur when the particles at fusion angles are located on the same sheet, and the equations have the same form as eqs. (32) and (33), with obvious replacement of particle indices to sheet-particle composite indices.

**Clustering** The cluster property affects only particles from the same sheet. For example, the two-particle form factor behaves as [75]

$$\lim_{\alpha \to \infty} F^{\mathcal{T}_n}_{\mu_1,\mu_2}(\theta_1 + \alpha, \theta_2) = \omega_n F^{\mathcal{T}_n}_{\mu_1} F^{\mathcal{T}_n}_{\mu_2}. \tag{39}$$

**Two-particle form factors** The two-particle form factors have special role. Based on the monodromy properties (36), one can define the minimal form factors, which has no pole in the physical sheet $\text{Im}(\theta) \in [0, \pi]$ as a solution to the monodromy equations:

$$F^{\mathcal{T}_n}_{\min|\mu_1,\mu_2}(\theta) = \hat{S}_{\mu_1\mu_2}(\theta) F^{\mathcal{T}_n}_{\min|\mu_2,\mu_1}(-\theta) = F^{\mathcal{T}_n}_{\min|\mu_2,\hat{\mu}_1}(2\pi i - \theta) \tag{40}$$

and keep repeating the cyclic property, one can also write:

$$F^{\mathcal{T}_n}_{\min|(s,a_1),(s+k,a_2)}(\theta) = F^{\mathcal{T}_n}_{\min|(s',a_1),(s'+k,a_2)}(\theta) \tag{41}$$

$$F^{\mathcal{T}_n}_{\min|(1,a_1),(s,a_2)}(\theta) = F^{\mathcal{T}_n}_{\min|(1,a_1),(1,a_2)}(2\pi(s-1)i - \theta) \tag{42}$$

and reach the conclusions that *i)* $F^{\mathcal{T}_n}_{\min|(1,a_1),(1,a_2)}$ determines any minimal form factors of $a_1$ and $a_2$ and *ii)* there are no poles of $F^{\mathcal{T}_n}_{\min|(1,a_1),(1,a_2)}$ on the extended sheet $\text{Im}(\theta) \in [0, n\pi]$.

The general two-particle form factor solution reads as [41]:

$$F^{\mathcal{T}_n}_{(s_1,a_1),(s_2,a_2)}(\theta) = \frac{Q^{s_1 s_2}_{a_1 a_2}(\theta;n)}{2nK_{s_1 s_2}(\theta;n)^{\delta_{a_1 a_2}} \prod_{\alpha \in \mathcal{S}_{a_1 a_2}} \left(B_\alpha(\theta;n)^{i_\alpha} B_{1-\alpha}(\theta;n)^{j_\alpha}\right)^{\delta_{s_1 s_2}}} \tag{43}$$

$$\times \frac{F^{\mathcal{T}_n}_{\min|(s_1,a_1),(s_2,a_2)}(\theta;n)}{F^{\mathcal{T}_n}_{\min|(s_1,a_1),(s_2,a_2)}(i\pi;n)} \tag{44}$$

with

$$K_{s_1,s_2}(\theta;n) = \frac{\sinh\left(\frac{i\pi(1-2(s_1-s_2))-\theta}{2n}\right)\sinh\left(\frac{i\pi(1-2(s_1-s_2))+\theta}{2n}\right)}{\sin\frac{\pi}{n}}, \tag{45}$$

$$B_\alpha(\theta;n) = \sinh\left(\frac{i\pi\alpha - \theta}{2n}\right)\sinh\left(\frac{i\pi\alpha + \theta}{2n}\right) \tag{46}$$

and

$$\begin{aligned} i_\alpha = n+1, & \quad j_\alpha = n, & \text{if} & \quad p_\alpha = 2n+1; \\ i_\alpha = n, & \quad j_\alpha = n, & \text{if} & \quad p_\alpha = 2n, \end{aligned} \tag{47}$$

where $p_\alpha$ are the exponents of the block factors in the corresponding scattering amplitude given Appendix in A. The $K$ and $B$ functions handle the kinematical and dynamical pole equations. The $Q$ functions satisfy

$$Q^{11}_{a_1 a_2}(\theta;n) = Q^{11}_{a_1 a_2}(-\theta;n) = Q^{11}_{a_1 a_2}(-\theta + 2i\pi n;n), \tag{48}$$

therefore they are polynomials in $\cosh\frac{\theta}{n}$, moreover from the asymptotic of the form factors, they are of finite order.

The minimal form factors $F^{\mathcal{T}_n}_{\min|\mu_1,\mu_2}$ satisfy (41), hence the only important ones are:

$$F^{\mathcal{T}_n}_{\min|(1,a_1),(1,a_2)}(\theta;n) = f_{a_1 a_2}(\theta;n) \tag{49}$$

and

$$f_{a_1 a_2}(\theta;n) = \left(-i \sinh \frac{\theta}{2n}\right)^{\delta_{a_1 a_2}} \prod_{\alpha \in \mathcal{S}_{a_1 a_2}} (f_\alpha(\theta;n))^{p_\alpha} \tag{50}$$

$$f_\alpha(\theta;n) = \exp\left[2 \int_0^\infty dt \, \frac{\cosh\left((\alpha - \frac{1}{2})t\right)}{t \cosh \frac{t}{2} \sinh(nt)} \sin^2\left(\frac{it}{2}\left(n + \frac{i\theta}{\pi}\right)\right)\right]. \tag{51}$$

In order to further specify the solution, one needs to determine the coefficients of the $Q$ polynomials, which we present for a few cases in the $E_7$ model in the following subsection.

### 3.4 Branch-point twist field form factors in $E_7$

In this section, we solve the bootstrap equations presented above. We work only in the paramagnetic phase, and based on the discussion in 3.1, we consider form factors with $\mathbb{Z}_2$ even particle configurations up to two particles. In particular, we solve for the following two-particle branch-point twist field form factors in the $E_7$ model:

$$F^{\mathcal{T}_n}_{(1,1)(1,1)}, \; F^{\mathcal{T}_n}_{(1,1)(1,3)}, \; F^{\mathcal{T}_n}_{(1,2)(1,2)}, \; F^{\mathcal{T}_n}_{(1,2)(1,4)}, \; F^{\mathcal{T}_n}_{(1,3)(1,3)}, \tag{52}$$

for $n = 2, 3$ and $4$. This, in turn, will provide us with one-particle form factors. Note that we restrict ourselves to $s_1 = s_2 = 1$, i.e. both paricles are on sheet 1. Then we can use eqs. (36) to extend to other sheets. Throughout this section we closely follow [18, 75] and borrow the $\Delta$-theorem argument from [36].

First of all for the calculation we need to determine the $k$ degree of the $Q^{11}_{a_1 a_2}$ polynomial for all form factors, mentioned above. Here we need to calculate the asymptotic of every function, which appears in the two-particle form factor formula. So for $\theta \to \infty$ we have:

$$K_{11}(\theta;n) \underset{\theta \to \infty}{\longrightarrow} -\frac{1}{4}\left(\sin \frac{\pi}{n}\right)^{-1} e^{\frac{\theta}{n}},$$

$$B_\alpha(\theta;n) \underset{\theta \to \infty}{\longrightarrow} -\frac{1}{4} e^{\frac{\theta}{n}}$$

for any choice of $\alpha$. Finally, using the results of Appendix B.1, the minimal two-particle form factors have to following asymptotics:

$$f_{11}(\theta;n) \underset{\theta \to \infty}{\longrightarrow} e^{\frac{3\theta}{2n}} \prod_{\alpha=0,\frac{10}{18},\frac{2}{18}} \left(\frac{\mathcal{N}(\alpha;n)}{2i}\right)^{p_\alpha} \tag{53}$$

$$f_{13}(\theta;n) \underset{\theta \to \infty}{\longrightarrow} e^{\frac{3\theta}{2n}} \prod_{\alpha=\frac{14}{18},\frac{10}{18},\frac{6}{18}} \left(\frac{\mathcal{N}(\alpha;n)}{2i}\right)^{p_\alpha} \tag{54}$$

$$f_{22}(\theta;n) \underset{\theta \to \infty}{\longrightarrow} e^{\frac{2\theta}{n}} \prod_{\alpha=0,\frac{12}{18},\frac{8}{18},\frac{2}{18}} \left(\frac{\mathcal{N}(\alpha;n)}{2i}\right)^{p_\alpha} \tag{55}$$

$$f_{24}(\theta;n) \underset{\theta \to \infty}{\longrightarrow} e^{\frac{2\theta}{n}} \prod_{\alpha=\frac{14}{18},\frac{8}{18},\frac{6}{18}} \left(\frac{\mathcal{N}(\alpha;n)}{2i}\right)^{p_\alpha} \tag{56}$$

$$f_{33}(\theta;n) \underset{\theta \to \infty}{\longrightarrow} e^{\frac{7\theta}{2n}} \prod_{\alpha=0,\frac{14}{18},\frac{2}{18},\frac{8}{18},\frac{12}{18}} \left(\frac{\mathcal{N}(\alpha;n)}{2i}\right)^{p_\alpha}. \tag{57}$$

Now we are in a position to fix the asymptotics of the $Q^{11}_{a_1 a_2}$ polynomials:

$$
\begin{aligned}
\lim_{\theta \to \infty} F^{\mathcal{T}_n}_{(1,1)(1,1)}(\theta) &\sim Q^{11}_{11}(\theta; n) e^{\frac{-3\theta}{2n}} &\Rightarrow&\quad Q^{11}_{11}(\theta; n) \sim e^{\frac{\theta}{n}} \\
\lim_{\theta \to \infty} F^{\mathcal{T}_n}_{(1,1)(1,3)}(\theta) &\sim Q^{11}_{13}(\theta; n) e^{\frac{-3\theta}{2n}} &\Rightarrow&\quad Q^{11}_{13}(\theta; n) \sim e^{\frac{\theta}{n}} \\
\lim_{\theta \to \infty} F^{\mathcal{T}_n}_{(1,2)(1,2)}(\theta) &\sim Q^{11}_{22}(\theta; n) e^{\frac{-2\theta}{n}} &\Rightarrow&\quad Q^{11}_{22}(\theta; n) \sim e^{2\frac{\theta}{n}} \\
\lim_{\theta \to \infty} F^{\mathcal{T}_n}_{(1,2)(1,4)}(\theta) &\sim Q^{11}_{24}(\theta; n) e^{\frac{-2\theta}{n}} &\Rightarrow&\quad Q^{11}_{24}(\theta; n) \sim e^{2\frac{\theta}{n}} \\
\lim_{\theta \to \infty} F^{\mathcal{T}_n}_{(1,3)(1,3)}(\theta) &\sim Q^{11}_{33}(\theta; n) e^{\frac{-7\theta}{2n}} &\Rightarrow&\quad Q^{11}_{33}(\theta; n) \sim e^{3\frac{\theta}{n}}
\end{aligned}
\tag{58}
$$

The desired asymptotics leads to the following Ansatz for the $Q$ polynomials:

$$
Q^{11}_{11} = A_{11}(n) + B_{11}(n) \cosh \frac{\theta}{n}
\tag{59}
$$

$$
Q^{11}_{13} = A_{13}(n) + B_{13}(n) \cosh \frac{\theta}{n}
\tag{60}
$$

$$
Q^{11}_{22} = A_{22}(n) + B_{22}(n) \cosh \frac{\theta}{n} + C_{22}(n) \cosh^2 \frac{\theta}{n}
\tag{61}
$$

$$
Q^{11}_{24} = A_{24}(n) + B_{24}(n) \cosh \frac{\theta}{n} + C_{24}(n) \cosh^2 \frac{\theta}{n}
\tag{62}
$$

$$
Q^{11}_{33} = A_{33}(n) + B_{33}(n) \cosh \frac{\theta}{n} + C_{33}(n) \cosh^2 \frac{\theta}{n} + D_{33}(n) \cosh^3 \frac{\theta}{n} .
\tag{63}
$$

The asymptotic behavior renders the left hands side of the cluster equations for the two-particle form factors $F^{\mathcal{T}_n}_{(1,1)(1,1)}$, $F^{\mathcal{T}_n}_{(1,1)(1,3)}$ and $F^{\mathcal{T}_n}_{(1,3)(1,3)}$ to be zero, leading to vanishing one particle form factors. This, in turn, verifies our initial claim that the twist fields are even operators. The two remaining cluster equations are:

$$
\frac{4^3 \sin \frac{\pi}{n} C_{22}(n)}{2 i n f_{22}(i\pi; n)} \prod_{\alpha = \frac{12}{18}, \frac{8}{18}, \frac{2}{18}} \left( \frac{\mathcal{N}(\alpha; n)}{2i} \right)^{p_\alpha} = \omega_{22} \left( F^{\mathcal{T}_n}_{(1,2)} \right)^2
\tag{64}
$$

$$
\frac{4^3 C_{24}(n)}{2 n f_{24}(i\pi; n)} \prod_{\alpha = \frac{14}{18}, \frac{8}{18}, \frac{6}{18}} \left( \frac{\mathcal{N}(\alpha; n)}{2i} \right)^{p_\alpha} = \omega_{24} F^{\mathcal{T}_n}_{(1,2)} F^{\mathcal{T}_n}_{(1,4)},
\tag{65}
$$

where the $\omega_{22}$ and $\omega_{24}$ are arbitrary phase factors. Assuming that the form factors are real, they can be $\pm 1$. The self-consistency of the form factor equations determines the correct choice.

We have thus written the complete ansatz for the two-particle form factors, so all that remains is to solve the bootstrap equations. We can write 12 bound state equations, 6 double pole equations, 3 kinematical pole equations and altogether 18 unknown coefficients and one-particle form factors to determine. However, the equations are not independent (see also [36]). It also turns out that the additional two nontrivial cluster equations are only compatible with the solution with the choice $\omega_{22} = \omega_{24} = +1$. Due to the nonlinearity of the cluster equations, we obtained two solution sets and determined the correct solution using the $\Delta$-theorem. The steps used in the solution are summarized in Appendix B.2.

The results obtained for the one-particle form factors and the coefficients are shown in Table 4.

### 3.4.1 von Neumann entropy: $n \to 1$ limit

The von Neumann entropy is obtained in the limit $n \to 1$. To achieve this, the form factor equations are expanded as a Taylor series in powers of $n - 1$. From the definition of the von

| $n$ | 2 | 3 | 4 |
|---|---|---|---|
| $F_{(1,2)}^{\mathcal{T}_n}$ | -0.19659438 | -0.22840073 | -0.23919615 |
| $F_{(1,4)}^{\mathcal{T}_n}$ | 0.07045544 | 0.08703329 | 0.09293670 |
| $F_{(1,5)}^{\mathcal{T}_n}$ | -0.03484937 | -0.07411714 | -0.04846618 |
| $F_{(1,7)}^{\mathcal{T}_n}$ | 0.00567573 | 0.01459751 | 0.00862978 |
| $A_{11}$ | 0.15825556 | 0.04136886 | 0.01439665 |
| $B_{11}$ | 0 | $-1.33190644 \, 10^{-10}$ | $2.73359941 \, 10^{-10}$ |
| $A_{13}$ | -0.09009461 | -0.01202218 | -0.00595536 |
| $B_{13}$ | 1.79108348 | -0.02385928 | $-1.19014979 \, 10^{-10}$ |
| $A_{22}$ | 0.04715041 | 0.00816131 | 0.00350597 |
| $B_{22}$ | 0.00273402 | -0.00697655 | -0.00613243 |
| $C_{22}$ | 0.01266897 | 0.00713472 | 0.00433196 |
| $A_{24}$ | -0.02996213 | -0.00318871 | -0.0013034 |
| $B_{24}$ | -0.00123958 | 0.001212597 | 0.00193404 |
| $C_{24}$ | -0.0052257 | -0.00556283 | -0.0013351 |
| $A_{33}$ | 0.00112214 | 0.000044664 | 0.00001498 |
| $B_{33}$ | 0.00069921 | -0.00006892 | -0.00004061 |
| $C_{33}$ | 0.0036519 | -0.00010010 | 0.00002939 |
| $D_{33}$ | -3.98790662 | 0.00029595 | $-1.18318365 \, 10^{-11}$ |

Table 4: The solutions of the form factors of $\mathcal{T}_n$ in case $n = 2$, 3 and 4.

Neumann entropy, we expect that the one-particle twist field form factors begin with the first power of $n - 1$, $F_{(1,a)}^{\mathcal{T}_n} \sim \mathcal{O}(n-1)$, where $a = 2, 4, 5, 7$. Consequently, the two previously written cluster equations imply that $C_{22} \sim \mathcal{O}((n-1)^2)$ and $C_{24} \sim \mathcal{O}((n-1)^2)$.

The calculation is very similar, however, during the solution, we retain only the leading terms, which eliminates the need to use the cluster equations. Thus, in this case, we solve the remaining 21 equations. The steps of the solution are relegated to Appendix B.2, shown in Table 12. The results of the computation are summarized in Table 5.

To compute the time evolution of the von Neumann entropy (cf. 4.1.2), we need the values of

$$g_a = \lim_{n \to 1} \frac{F_{(1,a)}^{\mathcal{T}_n}}{1 - n}, \tag{66}$$

which are provided in Table 6.

### 3.4.2 $\Delta$-theorem

The sum rule of the $\Delta$-theorem provides us with a way to verify the form factor solutions. The $\Delta$-theorem gives the UV scaling dimension of scaling fields as an integral of the two-point function of the trace of the energy momentum tensor and the scaling field under consideration [74]. For the twist fields, it reads as:

$$\Delta^{\mathcal{T}_n} = -\frac{1}{2\langle \mathcal{T} \rangle_n} \int_0^\infty dr \, r \, \langle \Theta(r) \mathcal{T}(0) \rangle_n \,. \tag{67}$$

| $n$ | 1 |
|---|---|
| $F^{\mathcal{T}_n}_{(1,2)}$ | $-0.581165103887824(n-1)+\mathcal{O}((n-1)^2)$ |
| $F^{\mathcal{T}_n}_{(1,4)}$ | $0.11600141887457172(n-1)+\mathcal{O}((n-1)^2)$ |
| $F^{\mathcal{T}_n}_{(1,5)}$ | $-0.04119906195620388(n-1)+\mathcal{O}((n-1)^2)$ |
| $F^{\mathcal{T}_n}_{(1,7)}$ | $0.004064317839147642(n-1)+\mathcal{O}((n-1)^2)$ |
| $A_{11}$ | $0.40071713298811207+\mathcal{O}(n-1)$ |
| $B_{11}$ | $-2.478163489212107(n-1)+\mathcal{O}((n-1)^2)$ |
| $A_{13}$ | $-0.5922850478117786(n-1)+\mathcal{O}((n-1)^2)$ |
| $B_{13}$ | $\mathcal{O}((n-1)^2)$ |
| $A_{22}$ | $0.20678784974108444+\mathcal{O}(n-1)$ |
| $B_{22}$ | $0.06450555538987324+\mathcal{O}(n-1)$ |
| $C_{22}$ | $\mathcal{O}((n-1)^2)$ |
| $A_{24}$ | $-0.4032046620289771(n-1)+\mathcal{O}((n-1)^2)$ |
| $B_{24}$ | $-0.08209442858427292(n-1)+\mathcal{O}((n-1)^2)$ |
| $C_{24}$ | $\mathcal{O}((n-1)^2)$ |
| $A_{33}$ | $0.06863470696506006+\mathcal{O}(n-1)$ |
| $B_{33}$ | $0.09225524208943167+\mathcal{O}(n-1)$ |
| $C_{33}$ | $0.028778168717632918+\mathcal{O}(n-1)$ |
| $D_{33}$ | $-0.32411146586680045(n-1)+\mathcal{O}((n-1)^2)$ |

Table 5: Expansion of the form factor solutions of $\mathcal{T}_n$ around $n=1$.

| | |
|---|---|
| $g_2$ | $0.581165103887824$ |
| $g_4$ | $-0.11600141887457172$ |
| $g_5$ | $0.04119906195620388$ |
| $g_7$ | $-0.004064317839147642$ |

Table 6: The function (66) for the even particles.

One can insert a complete set of multi-particle states, and carry out the integration over $r$ to get [41]:

$$\Delta^{\mathcal{T}_n} = -\frac{1}{2\langle\mathcal{T}\rangle_n}\sum_{k=1}^{\infty}\sum_{\mu_1...\mu_k}\int_{-\infty}^{\infty}\cdots\int_{-\infty}^{\infty}\frac{d\theta_1\ldots d\theta_k}{k!(2\pi)^k\left(\sum_{i=1}^{k}m_i\cosh\theta_i\right)^2}$$
$$\times\tilde{F}^{\Theta}_{\mu_1...\mu_k}(\theta_1,\ldots,\theta_k)\left(\tilde{F}^{\mathcal{T}_n}_{\mu_1...\mu_k}(\theta_1,\ldots,\theta_k)\right)^*, \tag{68}$$

where $\mu_i \equiv (j_i, a_i)$ is a composite index defined earlier, which contains the sheet number and particle type, and $\tilde{F}^{\mathcal{O}} = \langle\mathcal{O}\rangle F^{\mathcal{O}}$ is the full form factor. Fortunately, the sum converges rapidly, and the more relevant the operator, the faster the convergence [72, 77]. Since two-particle contributions often suffice for accurate predictions, we expect our earlier form factor results to perform well.

Moreover, only those contributions will be nonzero where the particles are on the same sheet, otherwise,

$$F^{\Theta}_{\mu_1...\mu_k} = 0, \tag{69}$$

since the different copies are considered noninteracting [41]. Thus, it is sufficient to consider

| $n$ | 1 | 2 | 3 | 4 |
|---|---|---|---|---|
| $\Delta_2^{\mathcal{T}_n}$ | $0.053755(n-1)+\mathcal{O}((n-1)^2)$ | 0.036369 | 0.063378 | 0.088498 |
| $\Delta_4^{\mathcal{T}_n}$ | $0.002142(n-1)+\mathcal{O}((n-1)^2)$ | 0.002602 | 0.004821 | 0.006863 |
| $\Delta_5^{\mathcal{T}_n}$ | $0.000270(n-1)+\mathcal{O}((n-1)^2)$ | 0.000457 | 0.001458 | 0.001271 |
| $\Delta_7^{\mathcal{T}_n}$ | $0.000002(n-1)\mathcal{O}((n-1)^2)$ | 0.000007 | 0.000028 | 0.000022 |
| $\Delta_{11}^{\mathcal{T}_n}$ | $\mathcal{O}((n-1)^2)$ | 0.002440 | 0.004718 | 0.006815 |
| $\Delta_{13}^{\mathcal{T}_n}, \Delta_{31}^{\mathcal{T}_n}$ | $\mathcal{O}((n-1)^2)$ | 0.000272 | 0.001049 | 0.000825 |
| $\Delta_{22}^{\mathcal{T}_n}$ | $\mathcal{O}((n-1)^2)$ | 0.000654 | 0.001317 | 0.001928 |
| $\Delta_{24}^{\mathcal{T}_n}, \Delta_{42}^{\mathcal{T}_n}$ | $\mathcal{O}((n-1)^2)$ | 0.000081 | 0.000337 | 0.000255 |
| $\Delta_{33}^{\mathcal{T}_n}$ | $\mathcal{O}((n-1)^2)$ | 0.000024 | 0.000061 | 0.000016 |
| $\Delta^{\mathcal{T}_n}$ | $0.056170(n-1)+\mathcal{O}((n-1)^2)$ | 0.043278 | 0.078553 | 0.107572 |
| exact | $0.058333(n-1)+\mathcal{O}((n-1)^2)$ | 0.04375 | 0.077778 | 0.109375 |

Table 7: One- and two-particle contributions to the $\Delta$-theorem, together with the sum of the given contributions and the exact result $\Delta^{\mathcal{T}_n} = \frac{c}{24}\left(n - \frac{1}{n}\right)$ for $n = 1, 2, 3, 4$.

the $j = 1$ sheet and multiply the results by $n$, since

$$
\begin{aligned}
F^{\Theta}_{(j,a_1)...(j,a_k)} &= F^{\Theta}_{(1,a_1)...(1,a_k)} \\
F^{\mathcal{T}_n}_{(j,a_1)...(j,a_k)} &= F^{\mathcal{T}_n}_{(1,a_1)...(1,a_k)}
\end{aligned}
\tag{70}
$$

for $j = 2,\ldots,n$. The corresponding form factors of the stress energy tensor were computed in [35]. The one- and two-particle contributions included in the calculation are summarized in Table 7, where

$$
\begin{aligned}
\Delta_a^{\mathcal{T}_n} &= -\frac{n}{2\langle\mathcal{T}\rangle_n} \int_{-\infty}^{\infty} d\theta_a \, \frac{1}{2\pi(m_a\cosh\theta_a)^2} \tilde{F}^{\Theta}_{(1,a)} \left(\tilde{F}^{\mathcal{T}_n}_{(1,a)}\right)^* \\
\Delta_{ab}^{\mathcal{T}_n} &= -\frac{n}{2\langle\mathcal{T}\rangle_n} \int_{-\infty}^{\infty} d\theta_a \int_{-\infty}^{\infty} d\theta_b \, \frac{1}{2(2\pi)^2(m_a\cosh\theta_a + m_b\cosh\theta_b)^2} \\
&\quad \times \tilde{F}^{\Theta}_{(1,a)(1,b)}(\theta_a,\theta_b) \left(\tilde{F}^{\mathcal{T}_n}_{(1,a)(1,b)}(\theta_a,\theta_b)\right)^* .
\end{aligned}
\tag{71}
$$

We note that, in our case, the $\Delta$-theorem is not only to test the solution, but also helps select the correct solution set.

We also note that in [36] the two solutions to the cluster equations are related to the leading- and subleading order/disorder operator, which are twist fields related to the $\mathbb{Z}_2$ symmetry of the theory. It is an interesting question whether in the case of branch-point twist fields the other solution has any physical significance.

# 4  Quench spectroscopy and form factors

## 4.1  Perturbation theory for small quenches

### 4.1.1  One-point functions

After a small quench one can exploit the integrability of the pre-quench theory [32, 78] or of the post-quench theory [33, 34, 79] to obtain a perturbative expansion for the one-point functions. For a mass quench in an integrable theory, both approaches can be used. However as it was pointed out in [79, 80] the post-quench expansion leads to a better approximation of the post-quench dynamics. In this section we present formulas based on the post-quench expansion.

Quantum quenches are sudden changes in some parameters of the system. In particular, we suppose that a quantum system is prepared in the ground state of $H_{\mathrm{pre}}$ and at $t = 0$ one suddenly changes the Hamiltonian to $H_{\mathrm{post}}$. In this work, we consider extended quantum systems, with local Hamiltonian

$$H_{\mathrm{pre}} = \sum_i h_i, \tag{72}$$

where $h_i$ is localised around the site $i$. Moreover, we consider global quenches, where the post-quench Hamiltonian is given as

$$H_{\mathrm{post}} = H_{\mathrm{pre}} + \lambda \sum_i g_i, \tag{73}$$

where $g_i$ are also local terms. In this context global quench means that the energy density is changed uniformly throughout the full system.

In the scaling field theory, we have

$$H_{\mathrm{post}} = H_{\mathrm{pre}} + \delta_\lambda \int dx\, \Phi(x, 0). \tag{74}$$

We refer to both $g_i$ and $\Phi(x, 0)$ as the quench operator.

Alternatively, we can write the pre-quench Hamiltonian as a perturbation of the post-quench one:

$$H_{\mathrm{pre}} = H_{\mathrm{post}} - \delta_\lambda \int dx\, \Phi(x, 0). \tag{75}$$

One can then formally write the pre-quench ground state in the post-quench basis. Rayleigh–Schrödinger perturbation theory leads to the first order result [79]:

$$|\Omega\rangle_{\mathrm{pre}} = |\Omega\rangle_{\mathrm{post}} + \delta_\lambda \sum_{l \neq \Omega} \int dx\, \frac{{}^{\mathrm{post}}\langle l|\, \Phi(x, 0)\, |\Omega\rangle_{\mathrm{post}}}{E_l^{\mathrm{post}}} |l\rangle_{\mathrm{post}} + O(\delta_\lambda^2), \tag{76}$$

where the sum over $l$ is the sum over all multi-particle states in the post-quench theory, $E_l^{\mathrm{post}}$ is the post-quench energy of the state and $|\Omega\rangle_{\mathrm{pre/post}}$ denote the pre/post-quench ground states respectively. Evaluating the spatial integral then leads to:

$$|\Omega\rangle_{\mathrm{pre}} = |\Omega\rangle_{\mathrm{post}} + 2\pi \delta_\lambda \sum_{l \neq \Omega} \frac{\delta(p_l^{\mathrm{post}})}{E_l^{\mathrm{post}}} \left(\tilde{F}_l^\Phi\right)^* |l\rangle_{\mathrm{post}} + O(\delta_\lambda^2), \tag{77}$$

where the form factors of $\Phi$ are meant to be evaluated in the post-quench theory.

In particular we are interested in the time evolution of one-point functions of local operators after the quench

$$\langle \mathcal{O}(t) \rangle_{\mathrm{post}} = {}_{\mathrm{pre}}\langle \Omega|\, e^{iH_{\mathrm{post}}t} \mathcal{O} e^{-iH_{\mathrm{post}}t}\, |\Omega\rangle_{\mathrm{pre}}. \tag{78}$$

Keeping only one-particle contributions, we finally get:

$$\langle \mathcal{O}(t) \rangle_{\text{post}} \approx_{\text{post}} \langle \Omega | \mathcal{O}(0,0) | \Omega \rangle_{\text{post}} + \delta_\lambda \frac{1}{m_1^2} \sum_a \frac{2}{r_a^2} \tilde{F}_a^\Phi \tilde{F}_a^\mathcal{O} \cos(m_a t) \,, \tag{79}$$

where we omitted higher order corrections in $\delta_\lambda$. In the above formula $\tilde{F}$ means the full form factor in the post-quench theory, i.e. normalized to include the post-quench vacuum expectation value, $m_i$ are the masses of particles in the post-quench theory and $r_a = \frac{m_a}{m_1}$.

From now on we suppose that we consider so-called mass quenches of the following form:

$$H_{\text{pre}} \;\; = \;\; H_{\text{CFT}} + \lambda \int dx \, \Phi(x,0) \tag{80}$$

$$H_{\text{post}} \;\; = \;\; H_{\text{CFT}} + (\lambda + \delta_\lambda) \int dx \, \Phi(x,0). \tag{81}$$

Invoking (14) we can write:

$$_{\text{pre}}\langle \Omega | \mathcal{O}(0,0) | \Omega \rangle_{\text{pre}} \;\; = \;\; A_\mathcal{O} \lambda^{\frac{2h_\mathcal{O}}{2-2h_\Phi}} \tag{82}$$

$$_{\text{post}}\langle \Omega | \mathcal{O}(0,0) | \Omega \rangle_{\text{post}} \;\; = \;\; A_\mathcal{O} (\lambda + \delta_\lambda)^{\frac{2h_\mathcal{O}}{2-2h_\Phi}} \,, \tag{83}$$

which allows us to relate the two expectation values at leading order:

$$_{\text{post}}\langle \Omega | \mathcal{O}(0,0) | \Omega \rangle_{\text{post}} \approx A_\mathcal{O} \lambda^{\frac{2h_\mathcal{O}}{2-2h_\Phi}} \left( 1 + \frac{\delta_\lambda}{\lambda} \frac{2h_\mathcal{O}}{2-2h_p} \right) = \bar{\mathcal{O}} \left( 1 + \frac{\delta_\lambda}{\lambda} \frac{2h_\mathcal{O}}{2-2h_\Phi} \right) \,, \tag{84}$$

where we introduced the notation $\bar{\mathcal{O}}$ for the pre-quench VEV of the operator considered.

Finally, after some reordering, we get:

$$\frac{_{\text{pre}}\langle \Omega | \mathcal{O}(0,t) | \Omega \rangle_{\text{pre}} - \bar{\mathcal{O}}}{\bar{\mathcal{O}}} \approx \frac{\delta_\lambda}{\lambda} \left[ \frac{2h_\mathcal{O}}{2-2h_\Phi} + \lambda \frac{\bar{\Phi}}{m_1^2} \sum_a \frac{2}{r_a^2} F_a^\Phi F_a^\mathcal{O} \cos(m_a t) \right] \,. \tag{85}$$

The second term of (85) predicts oscillations with frequencies related to the masses of elementary particles in the theory. These are seemingly undamped, however, it is not clear that at large times higher order terms modify this, see e.g. [34].

Second order terms lead to other oscillations, related to mass differences of particles. We do not consider these here, however, they are clearly visible in the numerical power spectra in Figures 5 and 10.

We also note that in the $E_7$ theory, there are $\mathbb{Z}_2$ even and odd particles. Therefore, in (85), nonzero terms only appear when the $\mathbb{Z}_2$ parity of the quench operator is the same as the particle.

### 4.1.2  Entropies

The formula for one-point functions can be directly applied to branch-point twist fields:

$$\frac{\Delta \mathcal{T}_n}{\bar{\mathcal{T}}_n} \approx \frac{\delta_\lambda}{\lambda} \left[ \frac{2h_{\mathcal{T}_n}}{2-2h_\Phi} + n\lambda \frac{\bar{\Phi}}{m_1^2} \sum_a \frac{2}{r_a^2} F_a^\Phi F_a^{\mathcal{T}_n} \cos(m_a t) \right] \,. \tag{86}$$

From the definition of the Rényi entropies

$$S_n(t) = \frac{1}{1-n} \log \left( \epsilon^{\Delta \mathcal{T}_n} \, _{\text{pre}}\langle \Omega | \mathcal{T}_n(0,t) | \Omega \rangle_{\text{pre}} \right) \tag{87}$$

| | constant | simulation | theoretical value |
|---|---|---|---|
| $\sigma$, FM | $C^{\sigma,\epsilon}$ | 0.04053 | 0.04167 |
| | $C^\epsilon$ | -0.10282 | |
| $\epsilon$, FM | $C^{\epsilon,\epsilon}$ | 0.10301 | 0.11111 |
| | $C^\epsilon$ | -0.09909 | |
| $\epsilon$, PM | $C^{\epsilon,\epsilon}$ | 0.11030 | 0.11111 |
| | $C^\epsilon$ | -0.10320 | |
| $n=1$ | $C^{\mathcal{T}_1,\epsilon}$ | -0.06624 | -0.06482 |
| | $C^\epsilon$ | 0.11179 | |
| $n=2$ | $C^{\mathcal{T}_2,\epsilon}$ | 0.04957 | 0.04861 |
| | $C^\epsilon$ | 0.11033 | |
| $n=3$ | $C^{\mathcal{T}_3,\epsilon}$ | 0.08674 | 0.08642 |
| | $C^\epsilon$ | 0.10612 | |
| $n=4$ | $C^{\mathcal{T}_4,\epsilon}$ | 0.12164 | 0.12153 |
| | $C^\epsilon$ | 0.10022 | |

Table 8: Best fit parameters for one-particle contributions to time evolution of various quantities. Entropy computations were done only in the paramagnetic phase.

one obtains:

$$S_n(t)-S_n(0) = \frac{1}{1-n}\log\left(1+\frac{\Delta\mathcal{T}_n}{\bar{\mathcal{T}}_n}\right) = \frac{1}{1-n}\frac{\Delta\mathcal{T}_n}{\bar{\mathcal{T}}_n} + \mathcal{O}(\delta_\lambda^2).$$ (88)

Restricing to one-particle contributions, one finally obtains:

$$(1-n)\big(S_n(t)-S_n(0)\big) \approx \frac{\delta_\lambda}{\lambda}\left[\frac{2h_{\mathcal{T}_n}}{2-2h_\Phi} + n\lambda\frac{\bar{\Phi}}{m_1^2}\sum_a\frac{2}{r_a^2}F_a^\Phi F_a^{\mathcal{T}_n}\cos(m_a t)\right].$$ (89)

These quantities are rather easily available in matrix product based calculations, that we present in the following subsection.

## 4.2 Quench simulations in the scaling limit

In order to simulate the post-quench dynamics we implemented the iTEBD algorithm [27, 28] for the $E_7$ line of the Blume–Capel spin chain, defined in 2. During the simulations, we determined the ground state using $N_{\rm im} = 200000$ imaginary iTEBD time steps with $\delta t_{\rm im} = 0.0005i$, and then performed $N_{\rm re} = 100000$ real time steps with $\delta t_{\rm re} = 0.005$, from which we extrapolated the scaling limit. In both cases, we used a bond dimension of $\chi_{max} = 200$ and applied a fourth-order Trotter–Suzuki approximation. We simulated various quenches, for different $\lambda$ but fixed quench parameter $\delta_\lambda/\lambda = 0.05$. Then using the method explained in 2.4 we rescaled the time, to have $m_1 = 1$. To be able to extrapolate to the scaling limit, we approximated the time signals with interpolating functions, and on a pre-defined time-grid we extrapolated to the scaling limit.

For the theory side, we included one-particle contributions in the form

$$\frac{\langle\mathcal{O}(t)\rangle-\bar{\mathcal{O}}}{\bar{\mathcal{O}}} \approx \frac{\delta_\lambda}{\lambda}\left[C^{\mathcal{O},\epsilon} + C^\epsilon\sum_a\frac{2}{r_a^2}F_a^\epsilon F_a^{\mathcal{O}}\cos(m_a t)\right]$$ (90)

$$(1-n)(S_n(t)-S_n(0)) \approx \frac{\delta_\lambda}{\lambda}\left[C^{\mathcal{T}_n,\epsilon} + nC^\epsilon\sum_a\frac{2}{r_a^2}F_a^\epsilon F_a^{\mathcal{T}_n}\cos(m_a t)\right].$$ (91)

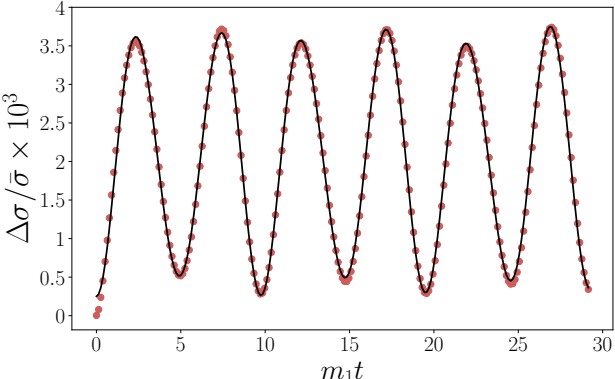

Figure 7: Time evolution of the leading magnetization after a quench in the ferromagnetic phase. The dots represent the extrapolated scaling limit obtained from the iTEBD simulations and the black line corresponds to (90) with best fit parameters $C^\epsilon$ and $C^{\sigma,\epsilon}$ given in Table 8.

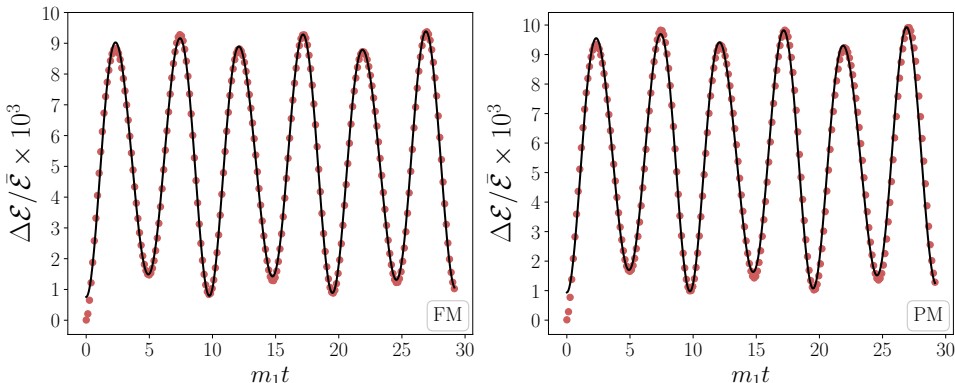

Figure 8: Time evolution of the quench operator after a quench in both phases. The dots represent the extrapolated scaling limit obtained from the iTEBD simulations and the black line correspond to (90) with best fit parameters $C^\epsilon$ and $C^{\epsilon,\epsilon}$ given in Table 8.

In order to compare our numerics, we used the form factors of the perturbing operators and the magnetic field computed in [35, 36], and the twist field form factors obtained in the previous section. The factors $C^\epsilon$ and $C^{Q,\epsilon}$ were obtained via a "best fit" choice to compare to the numerical results. We summarize them in Table 8. We note that $C^\epsilon$ we do not have a theoretical prediction, since it involves the relation between normalizations of the lattice operator and the limiting scaling field. However, it is important to mention that its value is roughly the same for each quantity, as expected.

### 4.2.1 One-point functions

First we carried out such calculations for the leading magnetic field and the perturbing operator.

The magnetic field couples to odd particles in the paramagnetic phase. On the contrary, the quench operator is even, therefore, we do not expect any time evolution of the magnetization. Indeed, our iTEBD simulations show no time evolution.

In the ferromagnetic phase, however, the magnetic field couples to even particles, and one expects a time evolution governed by (85). As it is shown in Figure 7, the numerical results

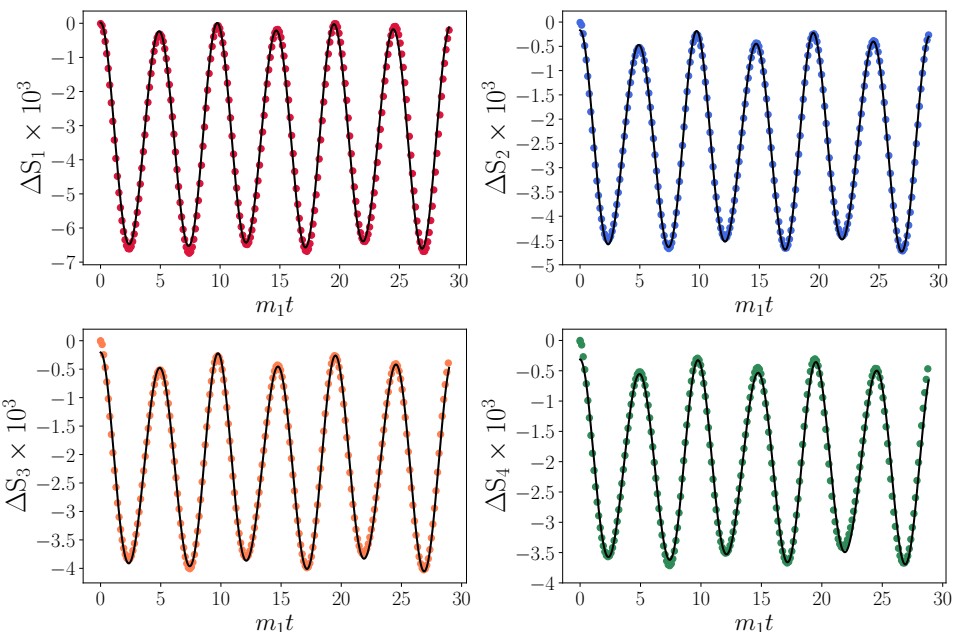

Figure 9: Time evolution of Rényi entropies together with (90) where the best fit parameter are summarized in Table 8.

match perfectly with our expectations. The best fit values are presented in Table 8.

Next we considered the one point function of the even perturbing (quench) operator, therefore it is expected to show a nontrivial time dependence in both the paramagnetic and ferromagnetic phases. Indeed, we can see in Figure 8. The agreement is excellent, using the best fit values presented in Table 8.

These results further confirm the validity of the form factor calculations of [35, 36].

### 4.2.2 Entropies

As it was mentioned earlier, in an $E_7$ mass quench only one-point functions of even operators have nontrivial time evolution. This is also true for the twist field one-point functions and the entropies derived from them. The first observation from our numerics is that the time evolution of Rényi entropies exhibit similar oscillating pattern as in the case of one-point functions, therefore our previous claim, that twist fields are even operators seems to be true. After the careful extrapolation process, we obtained our scaling field theory extrapolated results, which we present in Figure 9. The agreement with the best fit to the theory including one-particle contributions (i.e. eq. (89)) is excellent.

We note, that we run the iTEBD algorithm for quite long times containing $\approx 140$ periods related to the $m_2$, and we found no visible trace of linear entropy growth or damping of the one-particle oscillation, similarly to the $E_8$ case [18]. Moreover, using long-time data, we also extrapolated the power spectrum (i.e. quench spectroscopy), and found no trace of odd particles. The only visible peak forms at $m_4 - m_2$, the mass difference of the two lightest even particles. This belongs to the second-order term in the quench parameter mentioned earlier. We plot the power spectrum for an instance of lattice parameters in Figure 10, numerical mass ratios are the same as in Table 3.

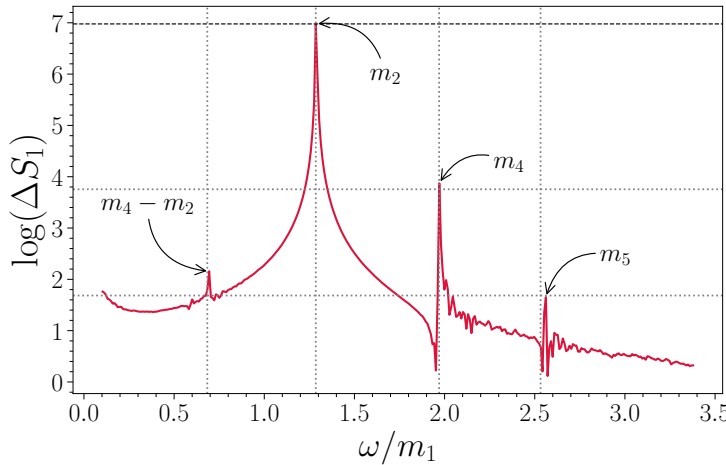

Figure 10: Quench spectroscopy for the von Neumann entropy in the Blume–Capel model on the $E_7$ line after the quench: $\lambda = 0.001$ and $\delta_\lambda = 0.00005$. Even particle contributions are visible. Horizontal lines correspond to $n \to 1$ limits of twist field and thermal operator form factors, the one related to $m_2$ is set by hand, and the other two were constructed using the form factors.

## 5  Outlook

In this work, we extended the form factor bootstrap approach to branch-point twist fields in the high-temperature phase in the $\mathbb{Z}_2$ even sector of the thermal deformation of the tricritical Ising model, the so-called $E_7$ scattering theory.

Energy density and leading magnetization form factors were used to approximate the time evolution of one-point functions. Time evolution of Rényi entropies is expressed in terms of branch-point twist field form factors. We implemented a numerical scaling limit procedure to the $E_7$ line of the Blume–Capel model. In a perturbative mass quench scenario, we studied the time evolution of one-point functions, Rényi and entanglement entropies in the time evolving state. For small quenches, we found an excellent agreement between the field theory prediction and the scaling limit of the spin chain model. We further verify that quench spectroscopy serves as a testing ground for form factor predictions. In an experimental realization of such a quench scenario, this would lead to a direct measurement protocol for form factors.

With this analysis, we barely scratched the surface of the richness of the Blume–Capel model. The first apparent question is the reliability of the "von Gehlen tricritical point", and the exact value of the tricritical couplings. This can be put in the context of non-invertible symmetries. The most evident question is, what is the Kramers–Wannier duality transformation of the model? To the best of our knowledge, this is not known in the literature. Looking for self-dual points, we could in principle, analytically find the line of tricritical fixed points. Furthermore, by analyzing the behavior of $\mathbb{Z}_2$ even operators under the duality, it would be possible to fix the operator corresponding to the $\epsilon$ field in the CFT. We already took some exploratory step to this direction, by analyzing the scaling limit of different combinations of even operators, looking for the one, which has approximately vanishing expectation value in the "von Gehlen tricritical point".

During this work, we only considered branch-point twist fields in the even sector. We are planning to examine the branch-point twist fields in the low-temperature phase. This would result in the construction of composite twist fields, see [69] for details of this problem in the Ising field theory. This would also allow to study entanglement assymetry and the quantum Mpemba effect [81] in the $E_7$ model

Another issue is related to the $\mathbb{Z}_2$ odd sector of the theory. One possible way is to study the dynamical structure factors, as they differ for even and odd operators [36]. This can be extracted using DMRG and TEBD to see whether one can trace back the odd particles [82][5]. Another way is to carry out quenches with an odd operator such as the leading or the subleading magnetic field.

Considering odd perturbations would lead to the "tricritical Ising spectroscopy" program, after the Ising spectroscopy by Zamolodchikov [83–85]. Turning on a magnetic field in the ferromagnetic phase leads to confinement of kink excitations [54] and the decay of the false vacuum [55]. The matrix product state simulation could be applied to study these phenomena in the Blume–Capel chain, as in the Ising and other spin-$\frac{1}{2}$ chains [15, 86] or more recently in the quantum Potts model [87]. The method was recently extended to the study of two-particle scatterings [88], which could also be carried out in the Blume–Capel case.

Finally, we mention that various experimental proposals can be found in the literature to realize the tricritical Ising fixed point [89–93]. We expect that findings of this work can be tested in experimental setups shortly. Our work also sheds light on the significance of proper scaling analysis of lattice results. This might prove important in for quantum simulations of different quantum field theories.

# Acknowledgements

We are indebted to Miklós Werner for sharing his insight into matrix product state calculations. We are grateful to Gábor Takács, Bence Fitos and István Csépányi for discussions. ML acknowledges countless fruitful discussions with István Szécsényi on branch-point twist fields and the scaling limit. The authors acknowledge funding from the Ministry of Culture and Innovation and the National Research, Development and Innovation Office (NKFIH) through the OTKA Grant K 134946. ML was supported by the Bolyai János Research Scholarship of the Hungarian Academy of Sciences. ML benefited from visits to SISSA Trieste, supported by the CNR/MTA Italy-Hungary 2023-2025 Joint Project "Effects of strong correlations in interacting many-body systems and quantum circuits". We also acknowledge KIFÜ (Governmental Agency for IT Development, Hungary) for awarding us access to the Komondor HPC facility based in Hungary.

# A   Particle spectrum and scattering in the $E_7$ model

In this appendix we present the necessary details of the integrable scattering of the $E_7$ theory [19, 20]. In Table 9 we present the masses and properties of the seven excitations of the theory. In Table 10 the scattering phases $S_{ab}$ between particles of type $a$ and $b$ are presented. Each scattering phase has the form of a product of blocks:

$$S_{ab}(\theta) = \prod_{\alpha \in \mathcal{A}_{ab}} (g_\alpha(\theta))^{p_\alpha}, \tag{A.1}$$

where

$$g_\alpha(\theta) \equiv \frac{\tanh \frac{1}{2}\left(\theta + i\pi\frac{\alpha}{18}\right)}{\tanh \frac{1}{2}\left(\theta - i\pi\frac{\alpha}{18}\right)}. \tag{A.2}$$

---

[5]iTEBD fails in this case, since translation invariance of the MPS ansatz is broken by operator insertions at different times.

In 10 ($\alpha$) means that the block $g_\alpha$ is present in the product (A.1) with $p_\alpha = 1$ and $(\alpha)^n$ is with $p_\alpha = n$. Bold superscripts indicate the particle type corresponding to the bound state pole. For two odd particles an extra $-$ sign should be included representing the parity of the particular excitations.

| exact | numerical | parity | excitation |
|---|---|---|---|
| $m_1$ | 1 | odd | kink |
| $m_2 = 2m_1 \cos \frac{5\pi}{18}$ | 1.285(6) | even | particle |
| $m_3 = 2m_1 \cos \frac{\pi}{9}$ | 1.879(4) | odd | kink |
| $m_4 = 2m_1 \cos \frac{\pi}{18}$ | 1.969(6) | even | particle |
| $m_5 = 4m_1 \cos \frac{\pi}{18} \cos \frac{5\pi}{18}$ | 2.532(1) | even | particle |
| $m_6 = 4m_1 \cos \frac{2\pi}{9} \cos \frac{\pi}{9}$ | 2.879(4) | odd | kink |
| $m_7 = 4m_1 \cos \frac{\pi}{18} \cos \frac{\pi}{9}$ | 3.701(7) | even | particle |

Table 9: Mass spectrum of the thermal deformation of the tricritical Ising model or $E_7$ model. Parity indicates the parity of the particle under $\mathbb{Z}_2$. The excitations in the high temperature phase are particles above the unique ground states. In the low-temperature phase they are either kinks extrapolating between the degenerate ground states or bound states of them, interpreted as particles above one of the vacua. The latter is indicated in the last column.

# B  Details of the twist field form factor calculation

## B.1  Two-particle form factor asymptotics

The $f_{ab}(\theta; n)$ minimal form factor can be written as a product of $f_\alpha$ blocks given in (50). With this, the minimal form factors come in the following form:

$$f_{11}(\theta; n) = -i \sinh \frac{\theta}{2n} \prod_{\alpha = \frac{10}{18}, \frac{2}{18}} f_\alpha(\theta; n)^{p_\alpha} \tag{B.1}$$

$$f_{13}(\theta; n) = \prod_{\alpha = \frac{14}{18}, \frac{10}{18}, \frac{6}{18}} f_\alpha(\theta; n)^{p_\alpha} \tag{B.2}$$

$$f_{22}(\theta; n) = -i \sinh \frac{\theta}{2n} \prod_{\alpha = \frac{12}{18}, \frac{8}{18}, \frac{2}{18}} f_\alpha(\theta; n)^{p_\alpha} \tag{B.3}$$

$$f_{24}(\theta; n) = \prod_{\alpha = \frac{14}{18}, \frac{8}{18}, \frac{6}{18}} f_\alpha(\theta; n)^{p_\alpha} \tag{B.4}$$

$$f_{33}(\theta; n) = -i \sinh \frac{\theta}{2n} \prod_{\alpha = \frac{14}{18}, \frac{2}{18}, \frac{8}{18}, \frac{12}{18}} f_\alpha(\theta; n)^{p_\alpha}. \tag{B.5}$$

Let us note, that

$$f_{\alpha=0}(\theta; n) = \exp \left\{ 2 \int_0^\infty \frac{dt}{t} \frac{\sin^2 \left[ \frac{t(i\pi n - \theta)}{2\pi} \right]}{\sinh(nt)} \right\} = -i \sinh \frac{\theta}{2n}. \tag{B.6}$$

| $a$ | $b$ | $S_{ab}$ | | $a$ | $b$ | $S_{ab}$ |
|---|---|---|---|---|---|---|
| 1 | 1 | $-\overset{2}{(10)}\overset{4}{(2)}$ | | 3 | 4 | $\overset{1}{(15)}(5)^2(7)^2(9)$ |
| 1 | 2 | $\overset{1}{(13)}\overset{3}{(7)}$ | | 3 | 5 | $\overset{1}{(16)}\overset{6}{(10)}{}^3(4)^2(6)^2$ |
| 1 | 3 | $-\overset{2}{(14)}\overset{4}{(10)}\overset{5}{(6)}$ | | 3 | 6 | $-\overset{2}{(16)}\overset{5}{(12)}{}^3\overset{7}{(8)}{}^3(4)^2$ |
| 1 | 4 | $\overset{1}{(17)}\overset{3}{(11)}\overset{6}{(3)}(9)$ | | 3 | 7 | $\overset{3}{(17)}\overset{6}{(13)}{}^3(3)^2(7)^4(9)^2$ |
| 1 | 5 | $\overset{3}{(14)}\overset{6}{(8)}(6)^2$ | | 4 | 4 | $\overset{4}{(12)}\overset{5}{(10)}{}^3\overset{7}{(4)}(2)^2$ |
| 1 | 6 | $-\overset{4}{(16)}\overset{5}{(12)}\overset{7}{(4)}(10)^2$ | | 4 | 5 | $\overset{2}{(15)}\overset{4}{(13)}{}^3\overset{7}{(7)}{}^3(9)$ |
| 1 | 7 | $\overset{6}{(15)}(9)(5)^2(7)^2$ | | 4 | 6 | $\overset{1}{(17)}\overset{6}{(11)}{}^3(3)^2(5)^2(9)^2$ |
| 2 | 2 | $\overset{2}{(12)}\overset{4}{(8)}\overset{5}{(2)}$ | | 4 | 7 | $\overset{4}{(16)}\overset{5}{(14)}{}^3(6)^4(8)^4$ |
| 2 | 3 | $\overset{1}{(15)}\overset{3}{(11)}\overset{6}{(5)}(9)$ | | 5 | 5 | $\overset{5}{(12)}{}^3(2)^2(4)^2(8)^4$ |
| 2 | 4 | $\overset{2}{(14)}\overset{5}{(8)}(6)^2$ | | 5 | 6 | $\overset{1}{(16)}\overset{3}{(14)}{}^3(6)^4(8)^4$ |
| 2 | 5 | $\overset{2}{(17)}\overset{4}{(13)}\overset{7}{(3)}(7)^2(9)$ | | 5 | 7 | $\overset{2}{(17)}\overset{4}{(15)}{}^3\overset{7}{(11)}{}^5(5)^4(9)^3$ |
| 2 | 6 | $\overset{3}{(15)}(7)^2(5)^2(9)$ | | 6 | 6 | $-\overset{4}{(14)}{}^3\overset{7}{(10)}{}^5(12)^4(16)^2$ |
| 2 | 7 | $\overset{5}{(16)}\overset{7}{(10)}{}^3(4)^2(6)^2$ | | 6 | 7 | $\overset{1}{(17)}\overset{3}{(15)}{}^3\overset{6}{(13)}{}^5(5)^6(9)^3$ |
| 3 | 3 | $-\overset{2}{(14)}\overset{7}{(2)}(8)^2(12)^2$ | | 7 | 7 | $\overset{2}{(16)}{}^3\overset{5}{(14)}{}^5\overset{7}{(12)}{}^7(8)^8$ |

Table 10: $S$-matrix amplitudes in the $E_7$ factorized scattering theory

Such a term can be factored out from $f_\alpha(\theta; n)$, which makes it easier to determine the asymptotics of the minimal form factors,

$$f_\alpha(\theta; n) = \exp\left\{2\int_0^\infty \frac{dt}{t}\frac{\sin^2\left[\frac{t(i\pi n-\theta)}{2\pi}\right]}{\sinh(nt)}\right\}\exp\left\{2\int_0^\infty\left[\frac{dt}{t}\frac{\cosh\left[t\left(\alpha-\frac{1}{2}\right)\right]}{\cosh\left(\frac{t}{2}\right)}-1\right]\frac{\sin^2\left[\frac{t(i\pi n-\theta)}{2\pi}\right]}{\sinh(nt)}\right\}.$$

By using some trigonometric identities one gets a simpler formula:

$$f_\alpha(\theta; n) = f_{\alpha=0}(\theta; n)\mathcal{N}(\alpha; n)\exp\left\{2\int_0^\infty\frac{dt}{t}\frac{\sinh\frac{t\alpha}{2}\sinh\left(\frac{t(1-\alpha)}{2}\right)\cos\left(\frac{t(i\pi n-\theta)}{\pi}\right)}{\cosh\frac{t}{2}\sinh(nt)}\right\} \quad\text{(B.7)}$$

with

$$\mathcal{N}(\alpha; n) = \exp\left\{-2\int_0^\infty\frac{dt}{t}\frac{\sinh\frac{t\alpha}{2}\sinh\left(\frac{t(1-\alpha)}{2}\right)}{\cosh\frac{t}{2}\sinh(nt)}\right\}. \quad\text{(B.8)}$$

The large $\theta$ asymptotics is then

$$\lim_{\theta\to\infty}f_\alpha(\theta; n) = \mathcal{N}(\alpha; n)\lim_{\theta\to\infty}f(\theta, 0; n) \sim \frac{\mathcal{N}(\alpha; n)}{2i}e^{\frac{\theta}{2n}}, \quad\text{(B.9)}$$

where $\mathcal{N}(\alpha; n)$ are convergent integrals.

## B.2 Steps of the twist field form factor evaluation

Table 11 for the case of $n = 2$, where the green dots indicate the equations used for verification. The solutions for the $n = 3, 4$ cases are similar, with the difference that steps 1, 2, and 3, and steps 7 and 8 merge because the kinematical pole equations will not only contain the $A_{ii}$ ($i = 1, 2, 3$) coefficients. Table 12 contains the procedure of the calculation in the $n \to 1$ case.

| $n = 2$ | | |
|---|---|---|
| **steps** | **used equations** | **defined unknowns** |
| 1. | kin. eq. for 11 | $A_{11}(n)$ |
| 2. | kin. eq. for 22 | $A_{22}(n)$ |
| 3. | $11 \to 2,\ 11 \to 4$ $22 \to 2,\ 22 \to 4$ cluster eq. for 22 | $F^{\mathcal{T}_n}_{(1,2)},\ F^{\mathcal{T}_n}_{(1,4)}$ $B_{11}(n)$ $B_{22}(n),\ C_{22}(n)$ |
| 4. | $22 \to 5$ | $F^{\mathcal{T}_n}_{(1,5)}$ |
| 5. | $13 \to 2,\ 13 \to 4$ | $A_{13}(n),\ B_{13}(n)$ |
| ● | $13 \to 5$ | |
| 6. | $24 \to 2,\ 24 \to 5$ $24 \to 13$ | $A_{24}(n),\ B_{24}(n)$ $C_{24}(n)$ |
| ● | cluster eq. for 24 | |
| ● | $24 \to 11$ | |
| 7. | kin. eq. for 33 | $A_{33}$ |
| 8. | $33 \to 2,\ 33 \to 7$ $33 \to 13,\ 33 \to 22$ | $F^{\mathcal{T}_n}_{(1,7)}$ $B_{33}(n),\ C_{33}(n),\ D_{33}(n)$ |
| ● | $33 \to 11$ | |
| ● | $33 \to 24$ | |

Table 11: The steps used to solve the bootstrap equations in the case $n = 2$.

| $n \to 1$ | | |
|---|---|---|
| **steps** | **used equations** | **defined unknowns** |
| 1. | $11 \to 2,\ 11 \to 4$ $22 \to 2,\ 22 \to 4$ kin. eq. for 11, kin. eq. for 22 | $F^{\mathcal{T}_n}_{(1,2)},\ F^{\mathcal{T}_n}_{(1,4)}$ $A_{11}(n),\ A_{22}(n)$ $B_{11}(n),\ B_{22}(n)$ |
| 2. | $22 \to 5$ | $F^{\mathcal{T}_n}_{(1,5)}$ |
| 3. | $13 \to 2,\ 13 \to 4$ | $A_{13}(n),\ B_{13}(n)$ |
| ● | $13 \to 5$ | |
| 4. | $24 \to 2,\ 24 \to 5$ | $A_{24}(n),\ B_{24}(n)$ |
| ● | $24 \to 13$ | |
| ● | $24 \to 11$ | |
| 5. | kin. eq. for 33 $33 \to 2,\ 33 \to 7$ $33 \to 13,\ 33 \to 22$ | $F^{\mathcal{T}_n}_{(1,7)}$ $A_{33},\ B_{33}(n)$ $C_{33}(n),\ D_{33}(n)$ |
| ● | $33 \to 11$ | |
| ● | $33 \to 24$ | |

Table 12: The steps used to solve the bootstrap equations in the case $n \to 1$.

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
