# Peer review of "Entanglement and quench dynamics in the thermally perturbed tricritical fixed point"

_SciPost Physics_

## Round 2 · Referee Report · Anonymous (Referee 1) · 2025-9-5

Strengths

  • The paper contains new, technically and numerically sound results for a highly non-trivial theory.
  • There is a successful comparison between quantum field theoretical predictions and numerical simulation on the lattice
  • New solutions to the form factor equations for the branch point twist field are found and tested for consistency
  • Overall, the physics of the E7 scattering theory is better understood and there are various suggested lines for further investigation.

Weaknesses

  • There are no major weaknesses in the analysis or results. My report below will suggest a few minor improvements around the following two areas.
  • Some further references that could be added
  • An improvement to the presentation of the section on entanglement entropy and twist fields

Report

In this paper, the authors carry out a comprehensive study of various models, depending on whether we look at the critical point, the lattice formulation or the quantum field theoretical picture. At criticality, they are dealing with the tricritical Ising model, in the lattice they are looking at the Blume-Capel model and in the continuous limit they are dealing with minimal Toda field theory associated with the E7 Lie algebra. The paper is generally well written and contains many new, interesting results. In my view it easily meets the publication standards of SciPost and should be accepted after minor corrections.

The main results of the paper are both of numerically and analytical nature, non-trivial in both cases. On the one hand, for the E7 theory, the authors use existing form factor calculations and carry out new ones (for the branch point twist field) in order to evaluate the post-quench one-point functions of these fields, using perturbation theory at first order in the quench parameter. Then they compare these findings with the corresponding one-point functions obtained numerically in the Blume-Capel model. Along the way, the results are tested in a number of ways, including via the Delta-sum rule. In particular, for the entanglement entropies, the authors find oscillatory behaviours at first order in perturbation theory, which are expected whenever the branch point twist field has non-vanishing one-point function.

Requested changes

I have a list of comments, some just about minor typos, others a little more conceptual. Here they are:

1) At the beginning of section 2.3 "Blume-Capen" should be "Blume-Capem"

2) On page 6, there are a couple of sentences I find hard to understand. "Note that to our knowledge the Kramers–Wannier duality transformation is not known for the Blume–Capel model, nevertheless, we keep with this analogy, and treat the results for the scaling limit of h⊥ as an operator as it is, and allow for a constant expectation value in the tricritical point. "

First, having "knowledge" and "known" in the same sentence sounds a bit repetitive, so one could rewrite this, but I don't know what is meant by treating the scaling limit "as it is". Also I think h⊥ is probably a capital H, if they are talking about the perturbing field in equation (7).

3) In the paragraph before equation (23) there is a typo "comlicated" should be "complicated". I also have a more conceptual comment. I don't think reference [39] used replica trick, in the sense that they don't consider n copies of the theory as is described here. Instead they see their fields as sitting at conical singularities in the complex plane and this geometry can be conformally mapped to an n-sheeted Riemann surface. The fact that they don't replica their theory is the reason why they get fields which are not the twist fields in equation (23). Their field Phi has dimension (24) divided by n. This is because their central charge is only c, while in the replica theory it is nc. For this reason, in the paper [39] it was claimed that the Renyi entropy is related to the nth power of the two-point function. This is different from equation (23). The replica picture that the authors use here was presented in this way in [41] instead. So, at the very least, they should cite both [39] and [41].

4) Before the conformal dimension in equation (24) the authors cite [39] where this dimension was not obtained (as I said, they have this divided by n). But also, the dimension of twist field in orbifolds was obtained much earlier in the 80s by people studying orbifold CFT such as in V.G. Knizhnik, Analytic fields on Riemann Surfaces II, Comm Math Phys 112, 567-590 (1987).

5) The sentence after figure (6) says that "In a massive theory the conformal invariance is broken, and the geometric argument fails". This is not entirely true (as the next sentence kind of says). What fails is that we can no longer conformally map the complex plane with punctures to the n-Riemann sheeted surface, but if we replicate our QFT we naturally generate a partition functions which lives in an n-sheeted Riemann surface, without the need for conformal invariance. So, in this sense, the geometric picture is still fine, and quite crucial.

6) After equation [52] the Delta-theorem is mentioned (I think for the first time in the paper?). The authors cite [36], which is fine, but if it is the first time it is mentioned, then it would make sense to cite the original paper here (their reference [74], which they cite later).

7) After equation [71] the authors have a sentence saying that the Delta-theorem helps select out the correct solution for the twist field. This is indeed, one of the main uses of the theorem. Later in the same paragraph they mention that it would be interesting to find out if the "other solution" for the twist field is meaningful.

I will write something about this here, in case it helps the authors make progress on this question and perhaps they can add some of this to their paper. It is generally natural and possible that they same twist field form factor equations have multiple valid, distinct solutions. This is particularly likely in complicated theories with a rich field content like the one here. Any composite twist field where the composition is between the branch point twist field and another neutral field (where neutral just means it is not another symmetry field), would satisfy the same branch point twist field equations as given here. The conformal dimension of such a field would be the dimension of the branch point twist field plus Delta/n, where Delta is the dimension of the neutral field. They could use Delta sum rule to test if their other solution is of this type. An example where two solutions are found and identified is the Lee-Yang theory, which was studied here https://arxiv.org/abs/1502.03275. See in particular equation (50) where two possible solutions for the one-point function are obtained.

8) The authors mention undamped oscillations after equation (85). There are by now quite a few examples where these are found. One further example that could be cited is for the sine-Gordon model, due to breather contributions which has non-vanishing one-particle form factors, see:
https://arxiv.org/pdf/2103.08492 and https://arxiv.org/pdf/2202.11113

9) I noticed that the authors refer to the E_7 scattering theory. I think many people will recognise this as the minimal Toda field theory associated with E_7. It might be worth mentioning this somewhere.

10) At the end of page 26 the authors mention the problem of solving the form factor equations for composite twist fields and cite [69] as an example. There are however other (and earlier!) examples of such solutions. For example: https://arxiv.org/abs/2105.13982 https://arxiv.org/abs/2301.01745

Recommendation

Publish (easily meets expectations and criteria for this Journal; among top 50%)

  • validity: high
  • significance: high
  • originality: high
  • clarity: high
  • formatting: excellent
  • grammar: excellent

Author:  Máté Lencsés  on 2025-11-05  [id 5998]

(in reply to Report 1 on 2025-09-05)

We are grateful to the anonymous Referee for the detailed and positive report, especially for the clarification on entanglement measures in quantum field theory and on the Delta sum rule.

Below we list the changes we made in the new version based on the report.

1.) At the beginning of section 2.3 ”Blume-Capen” is changed to ”Blume-Capel”

2.) We rephrased the sentence on page 6. to:

"To the best of our knowledge, the Kramers--Wannier duality transformation has not been established for the Blume--Capel model. Nevertheless, we keep with the Ising analogy, and treat the scaling limit of $h_i^{\perp}$ as the thermal perturbation, allowing for a nonzero expectation value in the tricritical point."

We also note, that capital $H^{\perp}$ in eq. (7) is the sum of local terms, denoted by lowercase $h_i^{\perp}$, see eq. (6). The scaling fields are then associated with local operators, therefore in this context it should be lowercase.

3,4,5)

We fixed the typo before equation (23).

We thank for the Referee for these important clarifications. We rephrased the paragraphs starting with the one just before eq. (23) to the one after eq. (24.). We fixed the references; now the replica trick is cited correctly (ref [41] in the previous, ref [44] in the new version). We also added a reference to the work of Knizhnik (ref. [73]).

6)

We rephrased the corresponding sentence to:

"Throughout this section we closely follow [18, 81]. To further specify the solution, we use the $\Delta$-theorem [80], similarly to [39]."

7)

We are grateful to the Referee for pointing this out. Indeed, the result of the delta theorem leads to a dimension which is compatible to the composite twist field, with the leading neutral operator, i.e. the thermal one. We added an extra subsection to the appendix with the numerical results (Appendix B.3.), and rephrased the corresponding paragraph in the main text (pg. 20.), and added ref. [84]

8)

We added these references to the introduction as refs. [19] and [20].

9)

Now we first mention the model as $E_7$ Toda field theory in the introduction.

10)

In view of Referee 2's report, we removed the speculation on composite twist fields from pg. 26. However, we extended the corresponding paragraph in the Outlook, where we added these references as refs. [91]
and [92].

---

## Round 2 · Referee Report · Anonymous (Referee 2) · 2025-9-19

Report

In this paper, the authors study an out-of-equilibrium protocol in the thermal perturbation of the tricritical Ising field theory and in the quantum Blume-Capel model, which provides a lattice realisation of the QFT.

The thermal perturbation of tricritical Ising, also known as the $E_7$ scattering theory, is an example of massive integrable QFT, with seven massive excitations. In this model, the authors study the evolution of the expectation value of local operators after a quantum quench, in particular, they study the expectation value of both the energy $\varepsilon$ and the twist operators $\mathcal{T}$ of the replicated theory, which yields the Rényi entropies. To study this protocol, the authors work in the limit of small quenches and use a perturbative expansion in the quench parameter in terms of form factor of the local operator of interest. To this avail, the authors use the already known results for the form factors of the field and they solve the bootstrap equation for the form factor of the twist fields $\mathcal{T}$.

The authors moreover study a lattice realisation of the QFT, the quantum Blume-Capel model. By performing a careful scaling limit, they identify the lattice operators corresponding to the magnetisation $\sigma$ in the field theory and, while they are unable to find the corresponding one for the energy field $\varepsilon$, they construct a field with the same scaling dimension as $\varepsilon$. The authors then perform a tensor network simulation of the mass quench protocol in the Blume-Capel model and compare the result with the field theory prediction, obtaining perfect agreement after a careful scaling limit. The authors observe that after the protocol the expectation value of the observables present persistent oscillations. By studying the Fourier transform of these expectation values (known as quench spectroscopy), they obtain clear peaks at frequencies corresponding to masses of the (even) particles in the $E_7$ scattering theory, confirming that the scaling limit of the lattice model is indeed given by this QFT.

The paper is well written and is very pedagogical, detailing every step needed to reproduce the result and to follow the computation. The paper is also very complete. The authors obtain several analytical results in a very non-trivial interacting field theory, and they complement them with a thorough numerical analysis of the corresponding lattice realisation. The study of out-of-equilibrium protocols in interacting field theories is a timely and interesting topic, with wide ramifications. For this reasons I fully recommend the publication in SciPost physics, after the authors answer some minor comments.

1- At page 12, in the context of the field theoretical computation of the entanglement entropies, the authors write

"The situation is different in the ferromagnetic phase, where there is a two-fold ground state degeneracy, and ground states transform nontrivially under the symmetry. However, different sectors are connected via a $\mathbb{Z}_2$ twist field (e.g. $\sigma$, the order parameter), and one has to consider composite twist fields. This was studied in the Ising model in [69]"

As the authors know, even in the ferromagnetic case the entanglement entropies are computed in terms of the standard twist fields, rather than the composite ones, and indeed in Ref. [69] the composite twist fields are used to compute the so called entanglement asymmetry. In fact, the composite twist fields are interesting also in the paramagnetic phase, where they are used to compute the symmetry resolved entanglement. I find the quoted paragraph potentially misleading for the non-expert reader, giving them the wrong impression that in the ferromagnetic phase the entanglement entropies are computed in terms of the composite twist field. I ask the authors to reformulate the paragraph to clarify this point.

2- On the topic of the entanglement asymmetry, I would like to bring to the attention of the authors the papers https://arxiv.org/abs/2008.11748 and https://arxiv.org/abs/1905.10487 by Casini, Huerta et al., which independently of Ref. [81] introduced a related quantity to the entanglement asymmetry under the name of entropic order parameter; I think that the authors should cite them together with [81].

3- This is optional, but I bring the paper https://arxiv.org/abs/2309.17199 to the attention of the authors. In this paper, it was studied the form factor bootstrap for both the standard and the composite twist fields in a different integrable deformation of the tricritical Ising CFT, the one with the vacancy field $t$. The authors may find it relevant.

Requested changes

1- Reformulate the paragraph

"The situation is different in the ferromagnetic phase, where there is a two-fold ground state degeneracy, and ground states transform nontrivially under the symmetry. However, different sectors are connected via a Z2 twist field (e.g. σ, the order parameter), and one has to consider composite twist fields. This was studied in the Ising model in [69]"

at pg. 12.

Recommendation

Ask for minor revision

  • validity: top
  • significance: high
  • originality: high
  • clarity: top
  • formatting: perfect
  • grammar: perfect

Author:  Máté Lencsés  on 2025-11-05  [id 5999]

(in reply to Report 2 on 2025-09-19)

We are grateful to the anonymous Referee for the positive report and the important and insightful comments.

Below we list the changes made in the new version in view of the report.

1)

Indeed the referee is completely right. What we meant is that the kink nature of the excitations should be considered during the computation of form factors, as it was considered in reference [75] ([69] in the previous version) for the Ising case. We rephrased the corresponding paragraph (last one starting on pg. 12), together
with the corresponding paragraph in the Outlook.

2)
We thank for the Referee for pointing out these references. We added them the discussion in the Outlook as refs. [94] and [95].

3)
We are grateful to the referee for pointing out this reference. We added a comment and ref. [56] in the end of Subsection 2.1.

---

## Round 2 · Referee Report · Anonymous (Referee 3) · 2025-10-6

Report

The authors consider the one-dimensional quantum Blume-Capel model in the scaling limit which realizes the thermal perturbation of the tricritical Ising model, which in turn is an integrable quantum field theory known as "E_7 theory" (with reference to its mass spectrum). They follow numerically the non-equilibrium dynamics produced by a homogeneous mass quench and compare with analytical results. The numerical results they obtain are of extremely good quality; the interpretation and presentation of the results need to be revised in view of the following facts:

  1. The authors are unaware that their numerical results fall within the theory of quantum quenches derived by Delfino and co-workers. This started in 2014 (Ref. [32]) with the first derivation of the post-quench state and one-point functions for quenches with interacting particle modes. This showed, in particular, why and when undamped oscillations of one-point functions are present up to a time-scale which goes to infinity as the size $\lambda$ of the quench is reduced (the result for the offset of the oscillations was obtained in Ref. [78]). It was explained in

[D20] Delfino G., Persistent oscillations after quantum quenches: The inhomogeneous case, Nuclear Physics B 954 (2020) 115002

why undamped oscillations are present in the full (non-perturbative) result if they are present at first order in perturbation theory. The theory was extended to generic dimension in

[DS22] Delfino G., Sorba M., Persistent oscillations after quantum quenches in d dimensions, Nuclear Physics B 974 (2022) 115643,

and the full non-perturbative structure was obtained in

[DS24] Delfino G., Sorba M., On unitary time evolution out of equilibrium, Nuclear Physics B 1005 (2024) 116587.

  1. The persistent oscillations which the authors observe numerically and emphasize in the abstract and in the introduction are predicted by Ref. [32] on the basis of the role (established in that paper) of the form factors of the quench operator and of the observable. In addition, Refs. [D20] and [DS24] explain why the oscillations which the authors observe numerically do not show any sign of decay even at large times.

  2. In section 4 the authors rewrite the perturbative formulae derived in Ref. [32] (and in Ref. [78] for the offset) replacing the states of the pre-quench basis with those of the post-quench one. The prosaic reason for the replacement is that as time increases the authors observe an improvement in the comparison with numerical results if the post-quench basis is used. The authors, however, run into a paradox when they try to theoretically justify the replacement saying that the two basis can both be used. Indeed, a perturbative expansion can be meaningfully performed only in the basis of the unperturbed theory, which for a perturbation in the quench size $\lambda$ is the pre-quench theory ($\lambda=0$). The paradox is clearly illustrated by Eq. (77), where the pre-quench state, which is the known initial condition of the dynamical problem, is written perturbatively in terms of the post-quench state, which is the unknown quantity to be determined.

What really happens is explained in Ref. [DS24]. There the analysis is non-perturbative and can (actually, must) be performed in the basis of the post-quench Hamiltonian which rules the dynamics. When the general expressions obtained in this way are specialized to the case of a small quench, the order $\lambda$ perturbative results of Refs. [32,78] are recovered. At this level, the pre-post replacement corresponds to a difference of order $\lambda^2$ and is mathematically immaterial. However, keeping the post-quench basis of the non-perturbative theory includes in the order $\lambda$ expressions contributions that in perturbation theory would be generated at higher orders, and this is the improvement observed in the comparison with numerics.

Recommendation

Ask for major revision

  • validity: -
  • significance: -
  • originality: -
  • clarity: -
  • formatting: -
  • grammar: -

Author:  Máté Lencsés  on 2025-11-05  [id 6000]

(in reply to Report 3 on 2025-10-06)

We thank Referee 3 for the comments. We are also grateful to the Referee for bringing our attention to refs. [D20] and [D22] and the recent development of ref. [DS24].

In view of the Referee's report, from a sentence in the introduction we eliminated the word "persistent", to put more emphasis that our claim is only valid for a limited time and we do not claim that the oscillations found are persistent on all time-scales. Now it reads as:

"Within the timescales accessible to our numerical simulations, the system exhibits undamped oscillations, while the corresponding entropies do not show the linear growth characteristic of thermalization."

We also added ref. [78] to the Introduction, together with the earlier ref [32]. Now these appear as refs. [34] and [35].

We also added a short historical introduction in the beginning of Section 4., where we also added ref. [DS24] as ref. [89], and also mention refs [D20] and [DS22] as refs [85] and [86], meanwhile, we replaced the title of Subsection 4.1. to "Post-quench perturbation theory for small quenches". We also eliminated the first paragraph of 4.1. since now it is redundant due to the new introduction. For the same reason, we also removed the sentence "These are seemingly undamped, however, it is not clear that at large times higher order terms modify this, see e.g. [34]", instead we mention the result of [34][35] and added references.

In the new introductory paragraphs, we summarize various approaches to the quench problem in quantum field theory and we argue why we use the perturbative computation in the post-quench basis. We completely agree with the Referee, that any viable computation should be done in the post-quench basis. This is indeed confirmed in ref. [57] where it was shown that the pre-quench perturbative calculation fails to predict the correct oscillation frequencies, even for a mass quench of 5\%, that is also our choice, while the computation in the post-quench basis leads to the correct frequencies.

---

## Round 3 · Author Response

We are grateful to the Referees for their constructive criticism and suggestions for improvement. Below we present the full list of changes in view of the referee reports. We also reply to the Referees separately below their report, explaining the changes made in detail.

---

## Round 3 · List of Changes

Pg.2. $E_7$ and $E_8$ scattering therioes are also mentioned as Toda field thoeries.
Pg.2 Added refs. [19] [20]
Pg.2. We removed the adjective: "persistent" from the sentence starting as "Within the timescales accesible to us..."
Pg.3. Ref [78] from the previous version is moved here, and now is ref. [35]
Pg.4 Comment on entanglemen along the massless perturbation and Ref.[56] added.
Pg. 4 Caption of Table 1. now contians our notation for the conformal weights.
In earlier version we used both $h_{\mathcal{O}}$ and $\Delta_{\mathcal{O}}$ to denote chiral conformal weights. In the new verions we use only $\Delta_{\mathcal{O}}$ to denote chiral weights throughout the paper.
Pg. 5 "Blume–Capem" is replaced to "Blume-Capel" in the beginning of 2.3
Pg. 6 Added ref [70]
Pg. 6 Last before paragraph of Subsec. 2.3 rephrased
Pg. 9 $\mathcal{E}$ is changed to $\epsilon$ in Table 2.
Pg. 11-12 Paragraphs rewritten starting from the one containing eq. (23) to the one after eq. (24)
Pg. 12-13 Rewritten paragraph starting on the bottom of Pg. 12.
Pg. 16 Sentence on the $\Delta$-theorem rewritten, reference [80] added.
Pg. 18 "$\cdot$" added in front of powers of $10$ in Table 4.
Pg. 20 Last paragraph of 3.4.2. rewritten, ref. [84] added.
Pg. 20-21 Historical introduction added in the beginning of Section 4., with addition of refs. [31],[85], [86] and [89].
Pg. 21 Subsection 4.1 is renamed to "Post-quench perturbation theory for small quenches"
Pg. 21 First paragraph of 4.1.1 removed.
Pg. 23 Paragraph after eq. (85) rewritten, refs [34][35] added
Pg. 27-28. Paragraphs reformulated, starting after: ".....expectation value in
the "von Gehlen tricritical point"". to "Considering odd perturbations...."; references [84],[91],[92],[94],[95] added
Pg. 33-34 Appendix B.3 on the composite twist field solution and the corresponding $\Delta$ sum rule added.
Pg.2 Added refs. [19] [20]
Pg.2. We removed the adjective: "persistent" from the sentence starting as "Within the timescales accesible to us..."
Pg.3. Ref [78] from the previous version is moved here, and now is ref. [35]
Pg.4 Comment on entanglemen along the massless perturbation and Ref.[56] added.
Pg. 4 Caption of Table 1. now contians our notation for the conformal weights.
In earlier version we used both $h_{\mathcal{O}}$ and $\Delta_{\mathcal{O}}$ to denote chiral conformal weights. In the new verions we use only $\Delta_{\mathcal{O}}$ to denote chiral weights throughout the paper.
Pg. 5 "Blume–Capem" is replaced to "Blume-Capel" in the beginning of 2.3
Pg. 6 Added ref [70]
Pg. 6 Last before paragraph of Subsec. 2.3 rephrased
Pg. 9 $\mathcal{E}$ is changed to $\epsilon$ in Table 2.
Pg. 11-12 Paragraphs rewritten starting from the one containing eq. (23) to the one after eq. (24)
Pg. 12-13 Rewritten paragraph starting on the bottom of Pg. 12.
Pg. 16 Sentence on the $\Delta$-theorem rewritten, reference [80] added.
Pg. 18 "$\cdot$" added in front of powers of $10$ in Table 4.
Pg. 20 Last paragraph of 3.4.2. rewritten, ref. [84] added.
Pg. 20-21 Historical introduction added in the beginning of Section 4., with addition of refs. [31],[85], [86] and [89].
Pg. 21 Subsection 4.1 is renamed to "Post-quench perturbation theory for small quenches"
Pg. 21 First paragraph of 4.1.1 removed.
Pg. 23 Paragraph after eq. (85) rewritten, refs [34][35] added
Pg. 27-28. Paragraphs reformulated, starting after: ".....expectation value in
the "von Gehlen tricritical point"". to "Considering odd perturbations...."; references [84],[91],[92],[94],[95] added
Pg. 33-34 Appendix B.3 on the composite twist field solution and the corresponding $\Delta$ sum rule added.

---

## Editorial Decision

resubmitted